# Sea level rise from West Antarctic mass loss significantly modified by large snowfall anomalies

Benjamin J. Davison [1] ✉, Anna E. Hogg [1], Richard Rigby [1], Sanne Veldhuijsen [2], Jan Melchior van Wessem [2], Michiel R. van den Broeke [2], Paul R. Holland[3], Heather L. Selley [1] & Pierre Dutrieux[3]

Mass loss from the West Antarctic Ice Sheet is dominated by glaciers draining into the Amundsen Sea Embayment (ASE), yet the impact of anomalous precipitation on the mass balance of the ASE is poorly known. Here we present a 25-year (1996–2021) record of ASE input-output mass balance and evaluate how two periods of anomalous precipitation affected its sea level contribution. Since 1996, the ASE has lost $3331 \pm 424$ Gt ice, contributing $9.2 \pm 1.2$ mm to global sea level. Overall, surface mass balance anomalies contributed little (7.7%) to total mass loss; however, two anomalous precipitation events had larger, albeit short-lived, impacts on rates of mass change. During 2009–2013, persistently low snowfall led to an additional $51 \pm 4$ Gt $yr^{-1}$ mass loss in those years (contributing positively to the total loss of $195 \pm 4$ Gt $yr^{-1}$). Contrastingly, extreme precipitation in the winters of 2019 and 2020 decreased mass loss by $60 \pm 16$ Gt $yr^{-1}$ during those years (contributing negatively to the total loss of $107 \pm 15$ Gt $yr^{-1}$). These results emphasise the important impact of extreme snowfall variability on the short-term sea level contribution from West Antarctica.

The glaciers draining into the Amundsen Sea Embayment (ASE) of West Antarctica are the dominant contributors to sea level rise from the Antarctic Ice Sheet[1–3]. Changes in the mass of these glaciers occur largely due to differences between net snow accumulation over the surface of each glacier basin and ice discharge into the ocean. In recent decades, the ASE glaciers have accelerated[4–7], thinned[8–10], and have undergone complex reconfigurations of their calving fronts and grounding lines[7,11,12] (Fig. 1). These ice dynamic changes have occurred asynchronously across the region[9] and have been interrupted by at least one period of stable or slightly declining speed around 2012[13,14].

The ice dynamic changes across the ASE have been attributed to periods of submarine melt-driven thinning of ice shelves in the region[15–19]. These periods of more rapid basal melting are driven by decadal-scale variations in the thickness of the warm Circumpolar Deep Water (CDW) layer[13,19], which is the main source of oceanic heat on the continental shelf[20]. These thickness variations are, in turn, thought to be driven largely by westerly wind anomalies at the shelf break[18,21–26], which causes an eastward undercurrent of CDW to divert southwards onto the continental shelf where it encounters cross-shelf troughs[21,27]. These westerly wind anomalies are correlated with sea surface temperature and air pressure anomalies in the tropical Pacific associated with the El Niño-Southern Oscillation (ENSO), the effects of which are transmitted to the Amundsen Sea via standing atmospheric Rossby waves[28,29] and manifest as a weakening and eastward shift of the Amundsen Sea Low (ASL) during El Niño events[30,31]. This ENSO-driven decadal variability in CDW thickness appears to be superimposed on a centennial trend of increasing westerly wind anomalies[18] and an associated increase in Amundsen Sea heat content[32]. While influential on

[1]School of Earth and Environment, University of Leeds, Leeds, UK. [2]Institute for Marine and Atmospheric Research Utrecht, Utrecht University, Utrecht, the Netherlands. [3]British Antarctic Survey, Cambridge, UK. ✉e-mail: b.davison@leeds.ac.uk

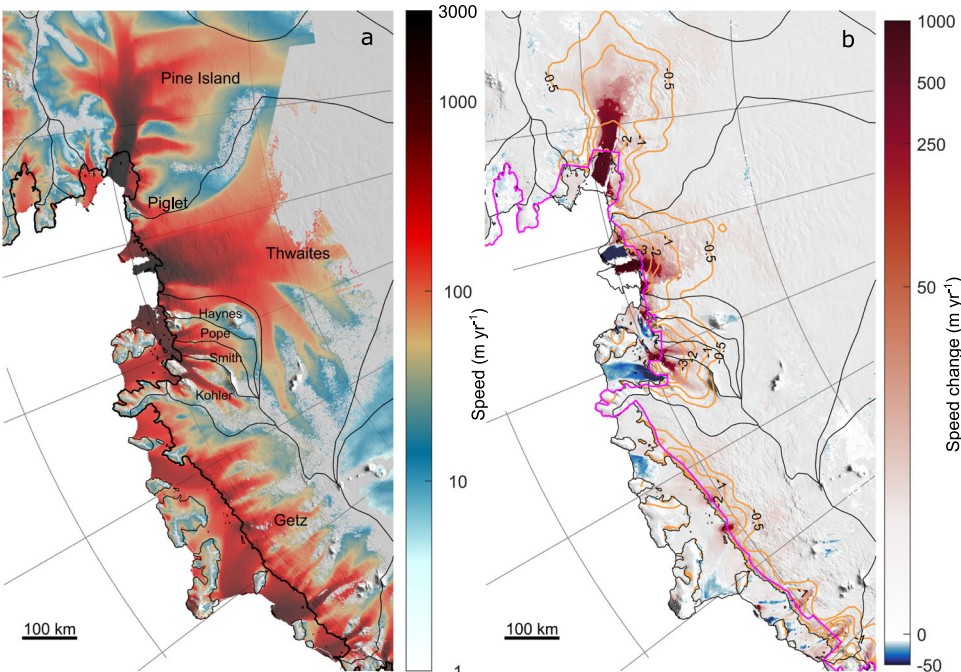

**Fig. 1 | Ice speed and speed change in the Amundsen Sea Embayment.**
**a** 2017–2021 ice speed over the Amundsen Sea Embayment derived from intensity tracking of Sentinel-1a and -1b Synthetic Aperture Radar images. **b** Ice speed change calculated as the difference between the mean 2021 speed from Sentinel-1 and the 2005–2008 MEaSUREs annual mosaics[62, 63]. Note the logarithmic colour scale used in both images. Mean ice thinning rates in m yr⁻¹ during 2003–2019[66] (orange contours), the 'FG1' flux gate[65] (magenta line), glacier drainage basins[100, 101] (thin black lines), and the ice-sheet grounding line[2] (thick black line), are also shown.

the wider Antarctic climate, the Southern Annular Mode (SAM) does not significantly influence winds directly over the Amundsen Sea, or rates of ocean melting at the grounding lines of the ASE ice streams that are rapidly thinning[18,33]. Overall, the observed ice dynamic changes of ASE glaciers in recent decades appear to be driven primarily by large-scale changes in atmospheric conditions, via their influence on surface winds, which drive cross-shelf exchange of warm ocean water masses.

Change in atmospheric conditions also drives significant precipitation variability over the ASE[34–37], which is the dominant contributor to surface mass balance (SMB) variations[35]. Most of the precipitation over the ASE is associated with synoptic-scale (~1000 km) weather systems that drive orographic precipitation near the coast[38]. In particular, variations in the strength and longitudinal position of the ASL and of anticyclones in the Bellingshausen Sea exert a strong influence on meridional wind and moisture fluxes[39]. In general, a deeper and more westward ASL, which is typical during winter, encourages the southward transport of maritime air masses and drives heavier precipitation over the ASE, particularly during the presence of a blocking anticyclone over the Bellingshausen Sea and Antarctic Peninsula, compared to times with a more eastward or weaker (i.e., higher pressure) ASL[30,31,37]. There are, therefore, seasonal fluctuations in the strength and position of the ASL, which typically cause heavier precipitation in the ASE during winter than in summer[30].

In addition to seasonal changes, the position and strength of the ASL are influenced by large-scale atmospheric circulation patterns and modes of climate variability. The ASL is deeper during positive phases of the SAM and during La Niña events[37,40–42], which encourages southward transport of warm, moist air, potentially forming so-called 'Atmospheric Rivers' (ARs)[43–45]. When these air masses make landfall, they can lead to short-lived but heavy orographic precipitation events[46]. Although infrequent and short-lived, extreme precipitation events (of which ARs drive a subset) contribute a large proportion of precipitation to the ASE and drive the majority of interannual

precipitation variability[34], and can cause notable (~18 cm) increases in ice-sheet surface height when integrated over a full winter season[47].

The impact of precipitation variations on mass balance has been studied extensively across the Antarctic Ice Sheet and over different time periods. Snowfall over the Antarctic Ice Sheet has increased throughout the 20th Century[48,49], but with substantial variability driven by southern hemisphere circulation patterns[37,50] and short-term variations driven by extreme precipitation events[35,36,51]. Over the 21st Century, increased SMB could even cause net ice-sheet mass gain under certain climate scenarios[52]. We note that these studies have largely focused on extreme increases in precipitation, and that comparatively little attention has been devoted to studying prolonged periods of low snowfall[53]. Despite the clear importance of atmospheric conditions in driving changes in CDW thickness and ice shelf basal melt rates, the direct effect on ASE mass balance of anomalous precipitation events, and their potential to mitigate or compound mass changes due to ocean-driven changes in grounding line flux from the ASE, has received comparatively little attention. This hinders our ability to determine the relative effects of changes in atmospheric and oceanic conditions on the mass balance of West Antarctica. Some recent observations provide evidence that such an analysis is pertinent. Firstly, a recent mass balance assessment of the Getz region noted that anomalously low precipitation during 2010/2011 winter exacerbated mass loss from the basin[5]. Secondly, anomalously high precipitation driven partly by ARs during the 2019/2020 winter caused rapid increases in surface height over the Amundsen Sea Embayment basins, despite ongoing dynamic thinning[47]. These observations indicate that periods of anomalously low or anomalously high precipitation may exert a non-negligible effect on the mass balance of the ASE.

Here, we present a 25-year-long record of ice flux and mass balance from West Antarctica, quantifying the sea-level contribution from this region from 1996 to 2021. Secondly, we quantify the effect of periods of anomalous precipitation on the mass balance of the ASE, and characterize the atmospheric conditions associated with those events in order to re-evaluate the impact of changing atmospheric

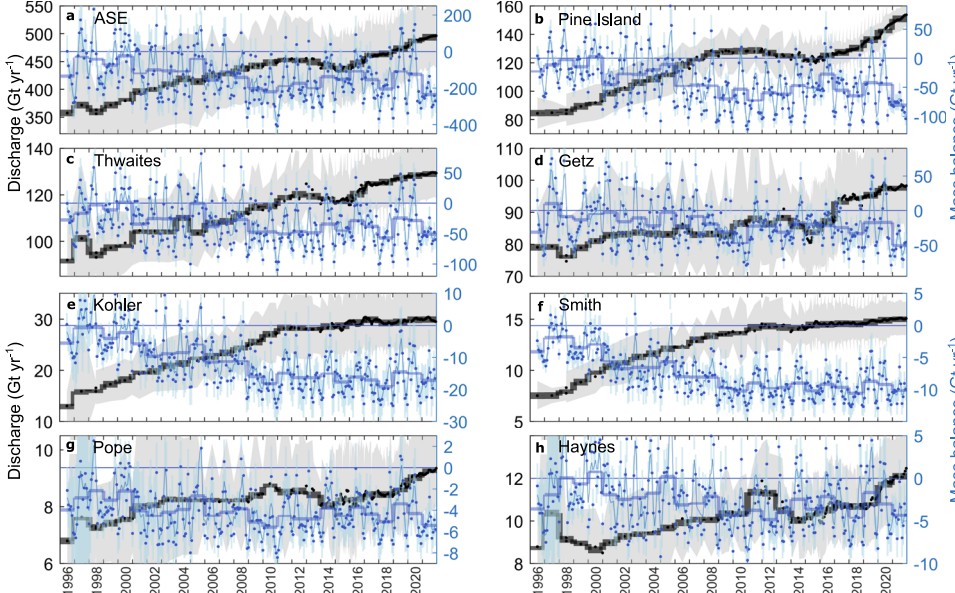

**Fig. 2 | Grounding line discharge and mass balance timeseries. a–h** Mass balance (blue) and grounding line discharge (black) from major Amundsen Sea Embayment basins. Individual observations are shown as filled markers with errors as vertical blue lines and grey shading. Annual averages are shown as semi-transparent horizontal bars and the 3-month rolling mean mass balance is given by the thin blue line.

conditions on the mass balance of ASE glaciers. To do this, we calculate grounding line discharge and mass balance for each glacier basin draining the ASE from 1996 to 2021 using existing and new velocity observations and three regional climate models (RACMO2.3p2[54], HIRHAM5[55], and MAR[56,57]). The synoptic-scale atmospheric conditions during periods of anomalous precipitation are evaluated using the Modern-Era Retrospective Analysis for Research and Applications, Version 2 (MERRA-2) global reanalysis dataset[58] and ERA5[59]. Finally, we examine historical records of climate indices, air pressure[60], and snow accumulation from ice cores[61] to assess the prevalence of, and atmospheric conditions associated with, comparable periods of anomalous precipitation in the 20th Century.

## Results

### Accelerating grounding line discharge

We combined publicly available, low-temporal resolution (3-monthly to annual) velocity estimates from 1996 to 2018[4,7,62–65] with high-resolution (6–12-day) velocity estimates from intensity tracking of Sentinel-1a and 1b image pairs since 2015 (Fig. 1a; Methods). These observations show widespread increases in speed across the ASE (Fig. 1b), continuing a trend observed since the early-1970s[4]. Acceleration is concentrated on the fast-flowing trunks of most major flow units in the study region (Fig. 1b), which has resulted in an increase in grounding line discharge (Fig. 2; Methods). Across the ASE, we estimate that grounding line discharge increased by 38.1%, from $357 \pm 63$ Gt yr$^{-1}$ in 1996 to $494 \pm 56$ Gt yr$^{-1}$ in 2021—an acceleration of $4.6 \pm 0.3$ Gt yr$^{-2}$ (Fig. 2a). Changes in discharge have been driven almost entirely by a widespread increase in speed across the region (Fig. 1b), which has offset the reduction (15.2 Gt yr$^{-1}$ or 3.1% in 2021) in grounding line discharge due to surface lowering[66] (Supplementary Fig. 1). Relative to 1996 values, the largest changes in discharge have occurred at Kohler Glacier (130% increase) and Smith Glacier (100%), whilst smaller relative changes have occurred at Getz (24%), Pope (36%), and Thwaites (40%) (Table 1).

Superimposed on this overall increase in discharge are variations in the rate and sign of change in discharge in each basin (Fig. 2a–c). Discharge from Pine Island Glacier increased episodically, with rapid increases in discharge from 1998 to 2009 (49.5%) and since late-2017 (18.6%). Separating these phases of rapidly increasing discharge was a

small (~5%) decline in discharge during 2012–2014 (Fig. 2b). This decline in discharge, detailed in ref. [14], is likely a delayed response to a reduction in cavity ocean heat content first observed in early 2012[13]. The reduced cavity ocean heat content was in turn driven in large part by lowering of the thermocline due to persistent easterly wind anomalies over the Amundsen Sea beginning in 2011, associated with a major La Niña event[13]. Building on previous observations[14], we observe this slight reduction in discharge across the easterly ASE basins, from Pine Island through to Kohler (Fig. 2b, c), suggesting that deepening of the thermocline was widespread on the eastern Amundsen Sea continental shelf. Discharge in these basins subsequently returned to ~2011 values and then stabilised until at least 2014. Grounding line discharge from the majority of basins has increased heterogeneously since 2014, most notably at Pine Island Glacier since 2017 potentially in response to several large calving events[7]. Furthermore, these calving events have led to the complete unbuttressing of the Pine Island Glacier southwest tributary (recently named 'Piglet Glacier') since 2016, which has subsequently accelerated by over 40% (Supplementary Fig. 1).

### Mass balance

Whilst grounding line discharge from the ASE glaciers has, other than a short hiatus around 2012, progressively increased since 1996, their mass balance has been more variable over the 25-year study period (Fig. 2d–f) because of large variations in SMB over seasonal to multi-year timescales. This observation holds regardless of which regional climate model is used (Supplementary Fig. 2). During the 1996/1997 hydrologic year (1st July 1996 to 30th June 1997), the ASE was likely in negative mass balance (−100 Gt yr$^{-1}$), though the errors are comparable to the mass balance at that time (Fig. 2d) and we have only one discharge estimate in each of 1996 and 1997. From 1996 to 2010, mass balance became increasingly negative, especially since 2006, as grounding line discharge increased rapidly and SMB remained relatively steady (Figs. 2 and 3b; Supplementary Fig. 3). Since 2010, the mass balance of the ASE and individual basins has remained negative and, apart from short-term fluctuations, relatively steady at $-168 \pm 78$ Gt yr$^{-1}$ on annual timescales (Fig. 2d–f). This relatively steady but negative annual mass balance since 2010 is partly due to the decline of grounding line discharge from 2012 to 2014, but is also due to large SMB anomalies, which we focus on here. We note that the mass

**Table 1 | Amundsen Sea Embayment grounding line discharge**

|  | PIG | | Thwaites | | Haynes | | Smith | | Pope | | Kohler | | Getz | | ASE | |
|---|---|---|---|---|---|---|---|---|---|---|---|---|---|---|---|---|
|  | Obs. | Err. | Obs. | Err. | Obs. | Err. | Obs. | Err. | Obs. | Err. | Obs. | Err. | Obs. | Err. | Obs. | Err. |
| 1996 | 84.4 | 11 | 92 | 14.4 | 8.8 | 1.5 | 7.5 | 1.4 | 6.8 | 1.3 | 13 | 2.7 | 79.1 | 11.5 | 357.3 | 11.5 |
| 1997 | 84.8 | 8.1 | 101.3 | 30.7 | 10.4 | 11.4 | 7.5 | 0.7 | 7.6 | 10.8 | 15.9 | 12.2 | 79.3 | 21.5 | 371.6 | 21.5 |
| 1998 | 85.5 | 8.1 | 94.1 | 10.8 | 9.1 | 1.2 | 8 | 1.1 | 7.3 | 1.1 | 16.2 | 2.7 | 75.6 | 11 | 359.5 | 11 |
| 1999 | 89.4 | 11.6 | 97.3 | 15.2 | 8.9 | 1.6 | 8.9 | 1.6 | 7.4 | 1.4 | 17.3 | 3.5 | 78.9 | 12.1 | 371.8 | 12.1 |
| 2000 | 92.4 | 12.4 | 98.6 | 10.7 | 8.6 | 1.4 | 9.7 | 1.6 | 7.6 | 1.8 | 18.2 | 3.6 | 80.9 | 14.3 | 378.4 | 14.3 |
| 2001 | 98.7 | 15 | 104.7 | 13.8 | 9 | 2.2 | 10.4 | 2 | 8 | 3.6 | 19.3 | 5.2 | 82.8 | 21.4 | 396 | 21.4 |
| 2002 | 102.5 | 13.6 | 104.4 | 13.4 | 9.3 | 1.7 | 10.8 | 2.2 | 8.1 | 2.6 | 19.8 | 4.1 | 83.2 | 14.5 | 401.5 | 14.5 |
| 2003 | 106 | 15 | 104.4 | 13.8 | 9.4 | 2 | 11.3 | 2.3 | 8.2 | 2.3 | 20.9 | 5.2 | 83.7 | 13.7 | 408.1 | 13.7 |
| 2004 | 108.5 | 14.7 | 110.2 | 19 | 9.5 | 1.7 | 11.6 | 2.3 | 8.2 | 2 | 21.4 | 5.5 | 83.3 | 12.7 | 418.8 | 12.7 |
| 2005 | 112.2 | 14.6 | 103.8 | 11.5 | 9.7 | 1.6 | 12.1 | 1.6 | 8.2 | 1.4 | 22 | 4.5 | 83.1 | 13.8 | 413.9 | 13.8 |
| 2006 | 118.2 | 14.1 | 107.9 | 12.4 | 9.9 | 1.5 | 12.4 | 1.7 | 8.2 | 1.4 | 22.7 | 4.3 | 85.2 | 14.6 | 425.8 | 14.6 |
| 2007 | 123.4 | 11.7 | 108.6 | 10.5 | 10.1 | 1.4 | 13 | 1.7 | 8.2 | 1.4 | 23.4 | 4.5 | 83 | 11.3 | 429.5 | 11.3 |
| 2008 | 126.6 | 11.5 | 111.7 | 10.8 | 10.1 | 1.4 | 13.4 | 1.8 | 8.2 | 1.4 | 24.5 | 4.7 | 83.2 | 13.9 | 437.4 | 13.9 |
| 2009 | 127.8 | 13.9 | 114.6 | 14.8 | 10.3 | 1.7 | 13.6 | 2.1 | 8.4 | 1.6 | 25.5 | 5.2 | 82.7 | 14.3 | 442.1 | 14.3 |
| 2010 | 128.2 | 13.8 | 115.2 | 13.3 | 10.3 | 1.5 | 13.7 | 2.3 | 8.7 | 1.8 | 27.1 | 5.4 | 86.5 | 17 | 447.7 | 17 |
| 2011 | 128.6 | 13.7 | 118.5 | 14 | 11.3 | 1.9 | 14.2 | 2.1 | 8.5 | 1.7 | 28.2 | 5.9 | 87.9 | 19.5 | 454.8 | 19.5 |
| 2012 | 126.8 | 14.8 | 120.2 | 15.5 | 11.3 | 2 | 14.3 | 2.6 | 8.6 | 1.7 | 28.2 | 6.2 | 86.4 | 15.1 | 453 | 15.1 |
| 2013 | 124.3 | 12.1 | 119.4 | 13 | 10.5 | 1.4 | 14.2 | 1.8 | 8.4 | 1.1 | 27.9 | 5.1 | 88.3 | 11.8 | 449.2 | 11.8 |
| 2014 | 123.7 | 12.9 | 117.7 | 13.2 | 10 | 1.6 | 14.1 | 1.8 | 8 | 1.5 | 28.5 | 5.4 | 85.6 | 12.2 | 441.7 | 12.2 |
| 2015 | 122.4 | 13.2 | 117.5 | 14.2 | 10.2 | 1.8 | 14.4 | 2 | 8.2 | 1.8 | 28.7 | 5.9 | 83.9 | 13.3 | 436.6 | 13.3 |
| 2016 | 124.7 | 11.7 | 120.3 | 13.7 | 10.5 | 1.8 | 14.5 | 2 | 8.2 | 1.7 | 29.3 | 6.1 | 87.2 | 13.4 | 445.9 | 13.4 |
| 2017 | 127.2 | 9.8 | 124.4 | 13.5 | 10.7 | 1.9 | 14.6 | 2.1 | 8.4 | 1.5 | 29.9 | 6.3 | 93.4 | 14.3 | 459.8 | 14.3 |
| 2018 | 130.3 | 9.5 | 125.4 | 13.2 | 10.7 | 1.7 | 14.5 | 1.9 | 8.4 | 1.4 | 29.3 | 5.5 | 94.1 | 13.6 | 463.7 | 13.6 |
| 2019 | 136.2 | 10.7 | 127.2 | 14 | 10.8 | 1.9 | 14.7 | 2 | 8.6 | 1.6 | 29.5 | 5.8 | 95.3 | 14.7 | 472.9 | 14.7 |
| 2020 | 143.2 | 10.6 | 128.3 | 14 | 11.7 | 2 | 14.9 | 2.1 | 9 | 1.6 | 29.9 | 6.2 | 97.6 | 15.4 | 485.1 | 15.4 |
| 2021 | 150.9 | 8.3 | 129.1 | 10.9 | 12.1 | 1.9 | 15 | 1.6 | 9.2 | 1.4 | 29.9 | 5.5 | 97.8 | 14.3 | 493.6 | 14.3 |

Mean annual (Jan–Dec) grounding line discharge (Gt yr⁻¹).

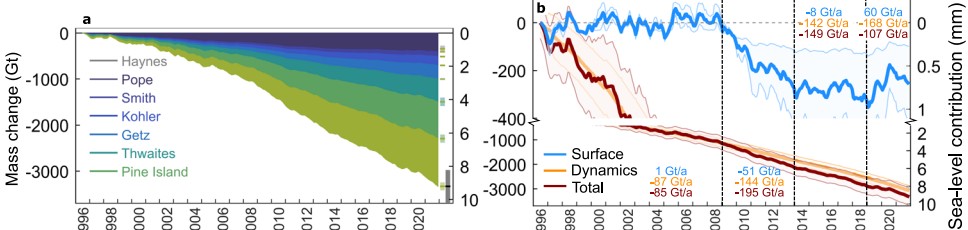

**Fig. 3 | Cumulative mass change since 1996. a** Cumulative mass change from major Amundsen Sea Embayment (ASE) basins, with sea-level equivalent shown on the right y-axis. Uncertainties are shown as floating bars after 2021—the grey bar indicates the total mass change uncertainty. **b** Cumulative mass changes partitioned between ice dynamics and surface mass balance for the ASE (basins G-H and F-G combined), and overall mass changes during time periods delineated by the vertical dashed lines.

balance of the ASE glaciers in 2021 was the lowest on record at −235 ± 68 Gt yr⁻¹, due to both increasing grounding line discharge since late-2017 and low snowfall during the 2021 winter (Fig. 3b; Supplementary Fig. 3).

Annually, the ASE has remained in negative mass balance during the entirety of our study period (Fig. 2d–f) such that the ASE basins (including the Getz basin) have lost 3331 ± 424 Gt of ice since 1996, equivalent to 9.2 ± 1.2 mm of global sea-level rise (assuming 1/361.8 mm of sea-level rise per Gt of ice loss[67]) (Fig. 3). Most of this mass loss has come from Pine Island Glacier (31.2%), Thwaites Glacier (23.8%) and Getz (14.9%), with the remainder coming from Kohler (8.9%), Smith (5.8%), Pope (3.1%) and Haynes (2.0%) glaciers (Fig. 3a).

## Anomalous precipitation events

We focus on two large and opposing changes to SMB during our study period (Fig. 3b). The first of these is a 5-year period from 2009 to 2013 inclusive, where SMB was largely below the climatological mean, whilst the second is a period of anomalously high snowfall during the winters (JJA) of 2019 and 2020. Together, these two events account for almost all of the cumulative SMB anomaly during the study period and therefore the majority of the mass change due to surface processes. In the ASE, precipitation is the dominant contributor to SMB (it explains over 98% of the variability in SMB); the remainder is largely caused by sublimation, with a negligible contribution from runoff. Consequently, we focus this study on investigating precipitation variations.

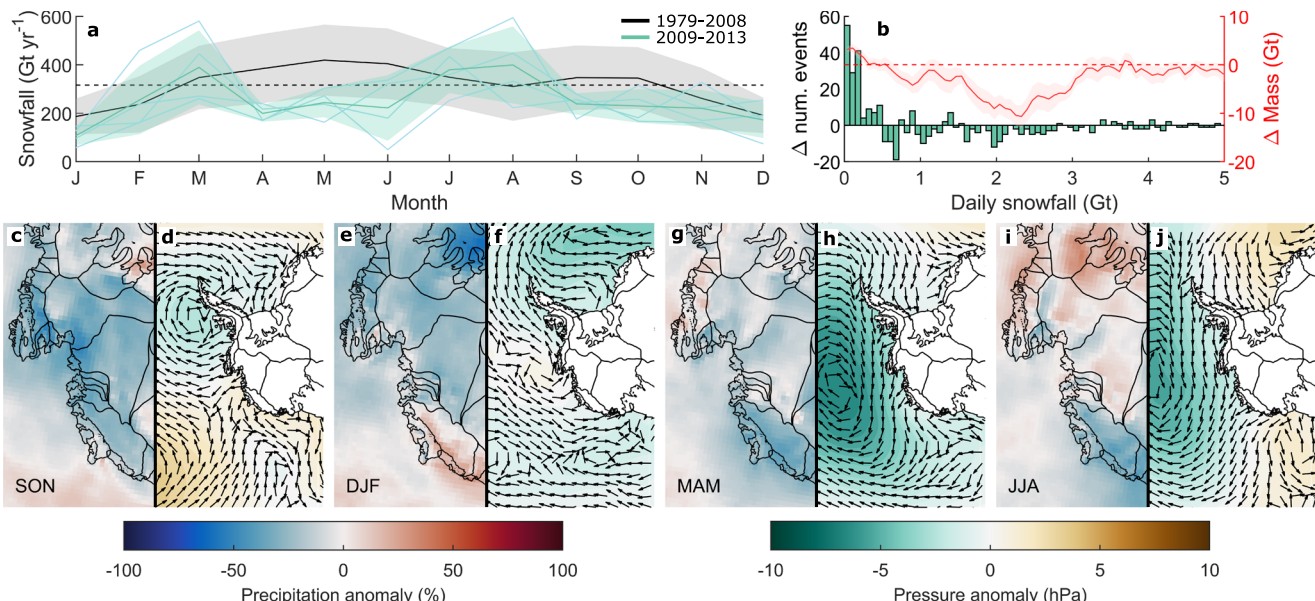

**Fig. 4 | Precipitation anomalies during 2009–2013. a** Monthly climatologies of RACMO2.3p2 precipitation during 1980–2008 (black line) with 1 s.d. (grey shading) and 2009–2013 (green line and shading). **b** Histogram of the change in the number of daily snowfall event sizes (green) and associated mass changes (red) during 2009–2013 compared to all other 5-year periods from 1980 to 2018. **c**–**j** Seasonal anomalies of monthly RACMO2.3p2 precipitation (**c, e, g, i**) and daily ERA5 surface air pressure and wind direction (**d, f, h, j**), relative to the 1980–2008 seasonal climatology. Antarctic coastline and drainage basins in **c**–**j** are also shown[100, 101] (black lines).

The anomalously low SMB during 2009–2013 (Figs. 3b and 4) has been identified previously in the Getz basin[5]; we provide further details here. Monthly RACMO2.3p2 precipitation during this time was, on average, just 17% below the monthly climatological mean. However, precipitation remained below the monthly climatological mean for 70% of the 2009–2013 period, creating a prolonged period of subdued precipitation (Fig. 4a). At daily timescales, the precipitation deficit was caused by a reduced number of large (-1.5–2.5 Gt) precipitation events during 2009–2013, compared to all other 5-year periods during 1979–2021 (Fig. 4b; Supplementary Fig. 4), consistent with fewer inferred AR events during this time period[44]. This persistently subdued precipitation led to a cumulative negative SMB anomaly that is unprecedented in all SMB datasets within the 1979–2021 period. There have been other periods of persistently below-normal precipitation, but none as prolonged as during 2009–2013 (Supplementary Figs. 3–5). We note that HIRHAM5 simulates more snowfall during this period compared to MAR and RACMO (Supplementary Fig. 3), but the spatial and temporal patterns of snowfall anomalies are similar between each model (Supplementary Figs. 3 and 4). Cumulatively, the 2009–2013 precipitation deficit caused 253 ± 81 Gt of additional mass loss across the ASE basins, averaging 51 ± 4 Gt yr⁻¹ during this period (Fig. 3b).

To investigate the cause of this period of anomalously low snowfall, we compare the spatial pattern of seasonal precipitation anomalies from each climate model (Supplementary Fig. 5) to surface pressure and wind direction anomalies from ERA5 (Fig. 4) and 6-hourly MERRA-2 reanalysis (Supplementary Fig. 6). In all climate models, ASE-integrated precipitation anomalies during 2009–2013 (relative to the monthly climatological mean) were most negative during the austral autumn and spring (Fig. 4; Supplementary Fig. 5). Precipitation was also anomalously low over the Getz basin during the 2009–2013 winters (Fig. 4i) and in the other ASE basins in each summer (Fig. 4e). Given that the greatest relative anomalies in snowfall also occurred during seasons of greatest absolute snowfall, then it follows that the greatest absolute snowfall anomalies were also in those seasons. Atmospheric circulation anomalies in both ERA5 and MERRA-2 were generally anomalously zonal or anomalously northward (Fig. 4d, f, h, j; Supplementary Fig. 6), thereby limiting the potential for moisture-bearing air masses to make landfall and induce orographic precipitation. These wind anomalies were associated with a range of surface air pressure conditions (Fig. 4c, e, g, i; Supplementary Fig. 6), but were generally characterised by a weaker ASL (for example, during the autumns of 2009–2013) or an anomalously eastwards ASL centre (as was more common during the winter and spring in 2009–2013). In addition, surface pressure in the Bellingshausen Sea and over the Antarctic Peninsula was anomalously low during most of 2009–2013, limiting the important blocking effect that anticyclones in that region typically provide[39]. These anomalous pressure patterns will have reduced the southward transport of maritime air masses to the coastline, as manifested in the anomalously zonal or northward wind anomalies (Fig. 4), thereby reducing orographic precipitation[30,31].

The second period of anomalous SMB occurred during 2019 and 2020 (Figs. 3 and 5). We identify large increases in SMB during the winters (JJA) of 2019 and 2020 that are associated with extreme precipitation events (Fig. 5; Supplementary Fig. 4). Precipitation across the ASE was 611 ± 33 Gt yr⁻¹ during the 2019 winter and 493 ± 257 Gt yr⁻¹ in the 2020 winter. Compared to the climatological mean of 355 ± 140 Gt yr⁻¹, precipitation was therefore 72% greater during the 2019 winter and 39% greater during the 2020 winter. A large proportion of this anomalous precipitation was caused by an increased number of high magnitude (>2.5 Gt) daily precipitation events (Fig. 5d; Supplementary Fig. 4). Indeed, global reanalysis data show that there were an anomalously high number of intense, meridional water vapour transport events (or ARs) (Methods) during both the 2019 and 2020 winters[45] (Fig. 5h–j), which temporarily increased ice-sheet surface height across the ASE in 2019[47]. Assuming these events delivered precipitation within the landfalling portion of the AR footprint during the AR detection period and in the 24 hours following the last detection[47,68], we estimate that 8.8% (14.9 Gt) and 8.7% (11.2 Gt) of the total winter precipitation during 2019 and 2020 respectively, was provided by ARs. By way of comparison, we also define 'extreme precipitation events' as days in the top 10% of long-term (1979–2021) daily precipitation totals[34] in RACMO2.3p2. By this metric, extreme precipitation events (timings in Fig. 5d) were more frequent during the

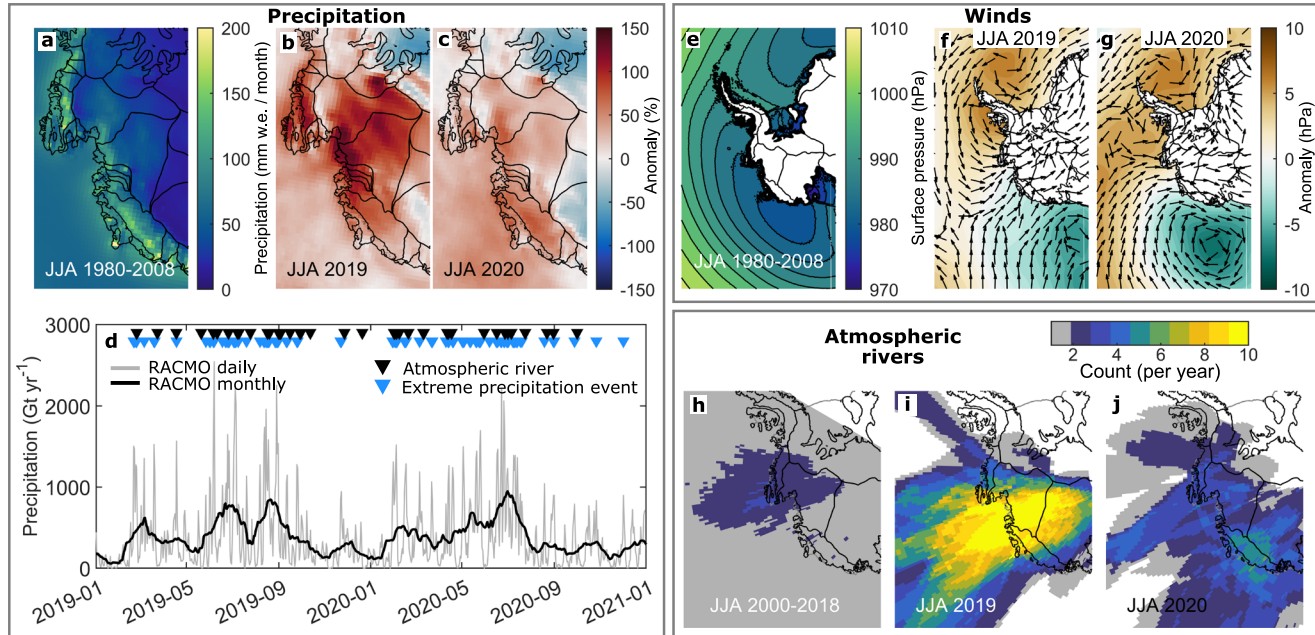

**Fig. 5 | Precipitation anomalies during 2019 and 2020. a** Winter (JJA) 1980–2008 climatology of monthly RACMO2.3p2 precipitation, and precipitation anomalies during **b** 2019 and **c** 2020. **d** Time-series of precipitation and timing of identified atmospheric rivers (black triangles) and extreme precipitation events (blue triangles) in 2019 and 2020. **e** JJA 1980–2008 climatology of daily ERA5 surface air pressure, and anomalies during **f** 2019 and **g** 2020, with ERA5 wind direction anomalies overlain (black arrows). **h–j** Count of cataloged atmospheric rivers during the winters of **h** 2000–2018, a period with few landfalling atmospheric rivers, **i** 2019, and **j** 2020. The coastline and major drainage basins are also shown[100,101] (black lines).

winters of 2019 and 2020 than in any other winter, and caused 39.3% (66.8 Gt) of the total winter precipitation in 2019 and 53.2% (69.4 Gt) in 2020.

In both the 2019 and 2020 winters, we identify a surface atmospheric pressure dipole in the ERA5 and MERRA-2 reanalyses, which is characterised by anomalously low surface pressure in the eastern Ross Sea and anomalously high pressure over the Antarctic Peninsula (Fig. 5f, g; Supplementary Fig. 7). This combination of a blocking high and an intense low-pressure system create a corridor for intense southward moisture transport from higher latitudes towards the Antarctic coastline, manifesting as northerly circulation anomalies and a higher frequency of intense orographic precipitation events (Fig. 5e–g; Supplementary Figs. 4 and 7). At times, atmospheric water vapour levels were sufficiently high to be classified as an AR (Methods), but there were many days or regions of high precipitation that were apparently not provided by an AR (Fig. 5d). This pressure dipole pattern is characteristic of ARs and extreme precipitation events around Antarctica[34,37,45] because such pressure dipoles are conducive to strong moisture convergence and southward transport of maritime air masses onto the Antarctic Ice Sheet[39].

## Discussion

The measurements presented here provide an update to the observational record of grounding line discharge and input-output mass balance of the ASE. From 1996 to 2021 inclusive, we estimate that the ASE has lost 3331 ± 424 Gt of ice, contributing 9.2 ± 1.2 mm to global sea level (Fig. 3). These discharge and mass change estimates are very similar to previous estimates[2,4,69] using equivalent methods in their overlapping time periods, with small differences owing primarily to the use of different bed topography and surface elevation data, but also to differences in velocity estimates and different approaches to filling data gaps.

Overall during 1996–2021, increases in grounding line discharge since 1996 have led to an additional mass loss of 3073 ± 496 Gt, contributing significantly to the total mass loss from the ASE. Therefore, 92.3% of the observed mass loss has been associated with a 38%

increase in grounding line discharge (Figs. 2a and 3b), with the remaining 7.7% due to cumulative anomalies in SMB since 1996 (Fig. 3b). Though we note that the ASE was not in balance during 1996 (Fig. 2), so it is not possible to estimate the precise contributions of SMB and discharge changes to mass loss including 1996. This relative split between changes in mass balance due to SMB and ice dynamics, often assumed to represent changes due to atmospheric and oceanic forcing, respectively, is nevertheless comparable to previous estimates[2,3,70]. Consequently, over the 25-year long study period, changes in grounding line discharge are the dominant driver of mass changes over the ASE.

The widespread increase in ice velocity across the ASE since at least the mid-1970s, particularly at Pine Island Glacier and Thwaites Glacier, has been the subject of many publications that have sought to constrain the oceanic and glaciological mechanisms responsible[7,13,14,19,26,71–73]. The ongoing speed-up of glaciers draining the ASE, and the commensurate increase in grounding line discharge, is consistent with the above studies and is the dominant signal during our study period. This speed-up appears to reflect variability in ocean temperature[19], with periods of speed-up occurring during inferred phases of relatively warm ocean temperature and periods of steady or slightly declining speed during phases of cooler ocean temperatures, as for example occurred around 2012[13,14] (Fig. 2a-c), though changes in damage, calving front position and grounding line location will have also contributed to the observed speed change, particularly towards the end of the study period[7,71,72].

Whilst acknowledging that this long-term speed up, likely in response to historical or recent oceanic forcing[17–19,74], is the dominant signal during our study period, the two periods of anomalous precipitation described in the results (Figs. 3–5) are less well-studied and so form the main focus of our discussion. The first of these is a period of anomalously low precipitation during 2009–2013 (Figs. 3b and 4) and the second is a period of anomalously high precipitation concentrated during the winters of 2019 and 2020 (Figs. 3b and 5). During those periods, changes in SMB (of which precipitation is the dominant term) had a substantial (26.3% during 2009–2013 and −56.3% during

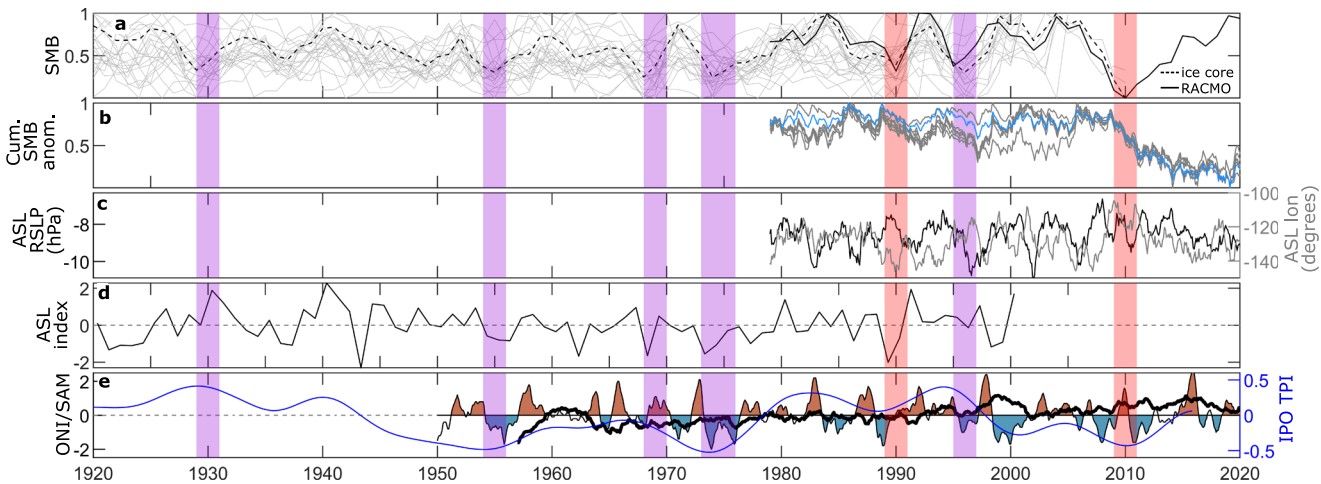

**Fig. 6 | Potential historical snowfall droughts. a** Normalised
RACMO2.3p2 surface mass balance (SMB; solid black line) and normalised snow
accumulation from 23 ice cores (grey lines; locations in Supplementary Fig. 9) and a
West Antarctic Ice-Sheet composite[49] (dashed black line). **b** Normalised RAC-
MO2.3p2 cumulative precipitation anomaly relative to the 1979–2008 mean, with
individual basins (grey lines) and the total Amundsen Sea Embayment (blue line)
shown. **c** Amundsen Sea Low (ASL) relative central pressure (black line) and centre
of longitude[60, 107] (grey line). **d** Detrended 20th Century ASL index[80], where a
negative index corresponds to a deeper ASL. **e** Timeseries of the Oceanic Niño
Index (ONI; blue and red shading), the Southern Annular Mode (SAM; thick black
line), and the Interdecadal Pacific Oscillation Tripole Index (IPOTPI; blue line). Red
and pink shading in all panels indicates periods of snowfall drought, based on
RACMO2.3p2 SMB and ice core snow accumulation timeseries, respectively
(Methods).

2019-2020) impact on total ASE mass loss during the respective time
periods. We stress, however, that these periods of anomalous pre-
cipitation were relatively short-lived, so their impact on total ASE mass
changes are small over longer timescales.

During 2009–2013, we characterise a period of prolonged
anomalously low SMB (henceforth 'snowfall drought') and quantify its
impact on the mass balance of the ASE. The cumulative mass loss due
to this snowfall drought was ~253 ± 81 Gt or 51 ± 4 Gt yr$^{-1}$ (Fig. 3b),
contributing positively to the total mass loss of 195 ± 4 Gt yr$^{-1}$. Com-
pared to periods of anomalously high precipitation, periods of
anomalously low precipitation such as this have received compara-
tively little research attention[34,35,37,45]. Across the Antarctic interior, an
important component of precipitation is frequent, light snowfall, often
termed 'clear-sky' precipitation[75,76]. Closer to the coast, however,
infrequent, short-lived and heavy precipitation events, driven by
intrusions of maritime air masses, dominate. These heavy precipitation
events drive most of the precipitation variability and the majority of
annual precipitation totals around much of coastal Antarctica[34,45]. The
snowfall drought during 2009–2013 was characterised by reduced
precipitation primarily in non-winter months for all basins excluding
Getz, and in non-summer months for Getz (Fig. 4), largely due to a
suppression of these relatively heavy precipitation events (Fig. 4b).
The suppression of precipitation was associated with both an anom-
alously weak or eastward ASL and an absence of a blocking anticyclone
in the Bellingshausen Sea, which resulted in zonal and northward wind
anomalies at the ASE coastline that limited the southward transport of
moist air[39,60,77] and therefore reduced orographic precipitation (Fig. 4).

Several modes of atmospheric variability can influence the
strength and position of the ASL[30,31], which exerts a strong control on
ASE precipitation variability. A relatively deep ASL is associated with
the La Niña phase of ENSO particularly during winter, with a positive
SAM during summer, and with a higher amplitude semi-annual
oscillation[30]. During 2009–2013, there was a switch to strong La Niña
conditions[17,19] and the SAM was positive (Fig. 6), which theoretically
should have deepened the ASL[30,31]. However, we observe the opposite
behaviour of the ASL, perhaps indicating that a suppression of the
semi-annual oscillation[78] may have been responsible for weakening the
ASL and reducing precipitation during 2009–2013. This inference is
consistent with the greater reduction in precipitation across the ASE
as a whole during spring and autumn (Fig. 4a).

The snowfall drought during 2009–2013 caused 51 ± 4 Gt yr$^{-1}$ of
relative mass loss in those years, accumulating to a total SMB anomaly
of −253 ± 81 Gt over the 5-year period (Fig. 3b). Grounding line dis-
charge during this time was approximately 450 Gt yr$^{-1}$, more than
double the estimated discharge during the mid-1970s[4]. Therefore, an
equivalent period of anomalously low precipitation during or before
the 1970s would have had a relatively greater influence on the mass
balance of the ASE. Indeed, based on discharge estimates for the mid-
1970s[4], a negative SMB anomaly of the magnitude we describe here is
equivalent to the cumulative mass loss caused by the observed
increase in grounding line discharge from 1974 to 1993 (Methods). In
other words, in terms of their total impact on sea-level rise, the 5-year
SMB anomaly we describe is equivalent to the observed 20-year
increase in discharge from the mid-1970s to the mid-1990s.

Given the potentially large impact of historical snowfall droughts
on the mass balance of the ASE, we examined historical records of
atmospheric conditions and snowfall accumulation, which may indi-
cate whether similar snowfall droughts have occurred in the past
century. Firstly, the timeseries of SMB from RACMO2.3p2 and snowfall
accumulation from ice cores (Fig. 6a) show that at least two other
multi-year periods of subdued snowfall occurred in the ASE since 1979
(red and pink shading in Fig. 6a), indicating that the snowfall drought
examined above was not a unique event. Although neither of these
earlier periods of drought are as prolonged or pronounced as that
during 2009–2013, there are similarities amongst these events in the
spatial patterns of surface air pressure (Supplementary Fig. 8) and
precipitation (Supplementary Figs. 9) anomalies, suggesting that they
share a common cause. In addition, each of those events occurred
during periods where the ASL central pressure was anomalously high
or located anomalously eastwards (Fig. 6c), providing further evidence
that these earlier snowfall droughts were similar to that during
2009–2013. Earlier in the 20th Century, ice core records (Fig. 6a;
locations in Supplementary Fig. 10) show that, although west Antarctic
snow accumulation has increased overall during the 20th Century[61,79],
there were several periods of relatively low snow accumulation around
1930 and in the mid-1950s, late-1960s and mid-1970s (pink shading in
Fig. 6). These are approximately coincident with the timing of inferred
ungrounding of Pine Island Ice Shelf from a bathymetric sill[74], the
retreat from which is ongoing, suggesting that historical snowfall
droughts may have contributed to these ice shelf geometry changes.

These fluctuations in snow accumulation recorded in ice cores do not appear to have a clear correspondence with a reconstructed ASL index[80] (Fig. 6d) or climate indices (Fig. 6e; Methods), complicating mechanistic interpretations. Overall, this evidence suggests that snowfall droughts may have occurred several times over the past century; however, future studies are required to confirm the existence of these droughts and to quantify their impact on the mass balance of the ASE.

We have shown that the 2009–2013 snowfall drought caused large changes in grounded ice mass over a 5-year period, significantly modifying the sea-level contribution from the ASE during the period of drought. However, the impact of this drought on the dynamics of the ASE is expected to be relatively small. To illustrate this, the ice-equivalent thinning due to the cumulative SMB anomaly from 2009–2013 is less than 50 cm (or 10 cm yr$^{-1}$) across the majority of the ASE basin, with the greatest relative thinning of -1.5 m concentrated in coastal parts of the Getz basin (Supplementary Fig. 11; Methods). This is at least an order of magnitude less than observed thinning rates near the grounding line of fast-flowing glaciers draining the ASE[66] (Fig. 1b). However, similar SMB-induced thinning in the 20$^{th}$ Century may have contributed to historical changes in ice dynamics, for example by altering the thickness of ice shelves, or the thickness and surface gradient of grounded ice, particularly if the SMB anomalies were more prolonged.

In contrast to the snowfall drought of 2009–2013, we identify anomalously high precipitation during the winters of 2019 and 2020. In total, 23.4 Gt (7.8%) and 136.2 Gt (45.3%) of the precipitation in the winters 2019 and 2020 were, according to our classification, caused by ARs and extreme precipitation events respectively. Extreme precipitation events therefore account for a large proportion of the positive winter precipitation anomaly (Supplementary Fig. 4). These events, particularly the ARs, are rare during other winters in our study period (Fig. 5h). Despite their high magnitude, this analysis suggests that ARs and extreme precipitation events have contributed little excess precipitation from 1996 to 2021.

Given the capacity of ARs to rapidly deliver large amounts of precipitation[47], we briefly discuss their capacity to increase snowfall supplies in the future and therefore potentially mitigate the future sea-level contribution from the ASE. Our results show that precipitation from landfalling ARs in the ASE (as detected using the *vIVT* method[45]) delivered less than 10% of precipitation during the extreme winters of 2019 and 2020. However, ARs are more common in East Antarctica[45], where they can contribute over 30% to the total precipitation in the region[68,81] and where they are expected to increase in frequency and intensity during the 21$^{st}$ Century[82,83] due to the greater moisture-bearing capacity of air masses as air temperature rises in future. On the other hand, ARs bring warm air to the ice-sheet that can cause intense surface melting[44,84] and could in the future cause liquid rather than solid precipitation and even SMB decreases by means of runoff, as currently occurs in Greenland[85,86]. To date, there are no dedicated projections of ARs or extreme precipitation events over Antarctica. However, most models project SMB increases over the grounded Antarctic Ice Sheet during the 21$^{st}$ Century[87], which may offset increases in grounding line discharge under certain climate scenarios[88], despite projected increases in low-elevation surface melting and associated meltwater runoff[89].

Over the past few decades, many studies have investigated the response of the ASE glaciers to changes in atmospheric and oceanic forcing[90], with a focus on processes driving cross-shelf exchange of warm CDW, ice-ocean interactions in ice shelf cavities and the resulting glacier response[7,13,14,17,19,26,71–73]. Simultaneously, many other studies have shown that substantial temporal variations in precipitation occur due to extreme precipitation events[37], and that variations in precipitation can affect the large-scale mass balance of the Antarctic Ice Sheet[35,36,50]. Here, we update the grounding line discharge and mass balance record from the ASE and use a range of atmospheric reanalysis datasets to combine these strands of research and re-evaluate the effect of anomalous precipitation on the mass balance of the ASE. We find that the ASE has lost 3331 ± 424 Gt ice, contributing 9.2 ± 1.2 mm to sea-level since 1996, driven by progressively increasing grounding line discharge. Overall during 1996–2021, changes in SMB had a modest (7.7%) impact on mass balance but anomalously high or low precipitation can have a larger (over 50%) impact on mass balance over shorter (2–5 year) timescales. Therefore, whilst discharge has progressively increased during the study period, the mass balance of the ASE has been comparatively steady since 2010. Thus, periods of anomalous precipitation do have a non-negligible impact on the sea-level contribution from the ASE, and estimates of mass balance over short time periods (5 years or less) may not be representative of longer-term mass changes. We suggest that further work is required to understand the effect of historical and future snowfall droughts and ARs on Antarctic Ice-Sheet mass change but that ice-ocean interactions remain a priority for future research as these are the dominant driver of ice dynamic variability and decadal-scale mass change in the ASE.

## Methods

### Grounding line discharge and ice velocity

We estimate grounding line discharge, $D$, across each flux gate pixel as

$$D = VHw\rho, \tag{1}$$

Where $V$ is the gate-normal ice velocity, $H$ is the ice-equivalent thickness, $w$ is the pixel width and $\rho$ is ice density (917 kg m$^{-3}$). We use "flux gate 1" (or FG1) from ref. [65] for our flux gate, which follows areas of direct bed observations and low bed elevation error, rather than the grounding line, and we resample the flux gate pixels to regular 200 m spacing. All plots in the main text are produced using this flux gate. We tested the sensitivity of our results to flux gate location by migrating each gate pixel along a flowline upstream based on the time-average velocity in 0.5-year increments for 10 years, creating 20 individual flux gates. Ice discharge increases by up to 6.3%, and by 1% on average, compared to that using the FG1 gate, as the gate is migrated upstream (Supplementary Fig. 12), reflecting the difficulty of conserving mass with imperfect thickness and velocity data, rather than algorithmic errors.

The gate-normal ice velocity is given by

$$\sin(\theta)V_x - \cos(\theta)V_y, \tag{2}$$

Where $V_x$ and $V_y$ are the easting and northing ice velocities, as defined by the South Polar Stereographic grid (EPSG3031), respectively, and $\theta$ is the angle of the flux gate relative to the same grid. Ice velocity data are compiled from multiple published and freely available sources during 1996–2018 and intensity tracking of Sentinel-1a and 1b images since 2015. For 1996 velocities, we use 450 × 450 m MEaSUREs InSAR-based estimates derived from 1-day repeat ERS-1 imagery[4,64], which covers the region spanning Cosgrove to Kohler Glacier, and 200×200 m velocities from ERS offset tracking over the Getz basin, provided by ENVEO (https://cryoportal.enveo.at/data/), which has been filled using an optimised ice-sheet model – BISICLES[5]. For 2000 and 2005 to 2016, we use 1×1 km MEaSUREs annual velocity mosaics[62,63]. We also use a MEaSUREs velocity mosaic incorporating velocity estimates between 1995 and 2001, at 450×450 m resolution[91]. For 1997–2018, we use 120×120 m ITSLIVE annual mosaics[65]; for 2009 to 2020, we use quarterly ice velocity mosaics from TerraSAR-X (2009–2016) and Sentinel-1 (2015–2020) just over Pine Island Glacier[7]. From 2015 onwards, we use high temporal resolution (6- to 12-day) estimates of ice velocity derived from intensity tracking of all 6- and 12-day Sentinel-1 image pairs in GAMMA, using standard methods[5]. We combine a range of cross-correlation window (ranging from ~1.2 × 1.2 km to ~500 × 500 m, each with 75% overlap), which are

merged using a signal-to-noise ratio weighted mean, to accommodate the large range in ice velocities in the ASE and to increase coverage. Individual image pair velocities are filtered using a 'dusting' approach to remove spatially-isolated pixels and a 2D gaussian filter[5], and posted at $100 \times 100$ m. Image pair velocities are mosaicked to $200 \times 200$ m monthly-mean velocities after further outlier removal and smoothing using a Discrete Cosine Transform Penalised Least Squares approach, optimised using the generalised cross-validation for fast, unsupervised processing[92,93]. Each of these velocity products spans a time period; following ref. [94], we ignore this and treat each product an instantaneous measurement with the timestamp given by the central date in the estimate. We extract easting and northing velocities at each gate pixel using nearest neighbour interpolation. Treating each gate pixel as a timeseries, we remove outliers, which we define as data points more than three standard deviations from the detrended timeseries. We fill temporal gaps using a linear interpolation except at the beginning and end of each timeseries, which are back- and forward-filled with the temporally nearest value for that pixel. Each filled timeseries (with no data gaps) is smoothed using a Savitzky-Golay filter with a 6-month window and a second-degree polynomial. Finally, as in previous studies[4,94], we assume the depth-averaged velocity is the same as the surface velocity.

We estimate ice thickness, and therefore discharge, at the timestamp of each velocity measurement using a fixed bed elevation and a time-varying ice surface. The assumption of a fixed bed means that we neglect any changes in bed elevation due to erosion or uplift, which we expect to be at least an order of magnitude smaller than the observed ice surface elevation changes. We use BedMachine version 2 bed topography[95,96] and the 200 m REMA Digital Elevation Model[97] (DEM) to define our baseline thickness estimate. We use observed ice-equivalent thinning rates from 2003–2019 derived from ICESat-1 and ICESat-2 observations[66] to adjust the REMA DEM surface, assuming that the REMA DEM is timestamped to 9th May 2015[97]. For velocity measurements prior to 2003, we use the average estimated elevation from 2003 to 2006, rather than extrapolate the thinning rate. Similarly, for velocity measurements after 2019, we use 2017–2019 average ice surface elevation. Our results are insensitive to these thickness adjustments (Supplementary Fig. 13). Ice-equivalent thickness is estimated at each velocity epoch by removing a time-varying firn air content, provided by the IMAU Firn Densification Model[98,99], forced by RACMO2.3p2[54] (Supplementary Fig. 14). Our flux gates are upstream of the grounding line; we remove the effect of dynamic thinning or thickening and SMB changes that occur between the flux gate and grounding line by integrating the observed ice-equivalent elevation change rates and modelled SMB between the flux gate and grounding line[100,101]. Relative to the final discharge estimate, these corrections are modest but non-negligible: 2.6% for dynamic thinning and 2.5% for SMB, across the full ASE flux gate. For all discharge estimates, we use the MEaSUREs grounding line[100,101], and so we ignore changes in grounding line position through time.

Grounding line discharge from each basin is estimated by integrating the pixel-based discharge estimate for all flux gate pixels that fall within the respective basin. In the main text, we use basins from ref. [100]. In this paper, we define the ASE as MEaSUREs basins G-H and F-G (i.e. we include the Getz basin in our ASE totals). Basin-scale discharge using alternative basin definitions[102] are shown in Supplementary Table 1.

## Discharge error
Following ref. [94], we define our discharge error using the minimum and maximum possible discharge based on the errors in the velocity and thickness data at each gate pixel. The pixel-based maximum discharge is defined as

$$D_{max} = (V + \sigma V)(H + \sigma H)w\rho, \qquad (3)$$

And the pixel-based minimum discharge is defined as

$$D_{min} = (V - \sigma V)(H - \sigma H)w\rho. \qquad (4)$$

$\sigma V$ and $\sigma H$ are the timestamped, pixel-based errors in the velocity and thickness estimates, respectively. Where available, we use the easting and northing velocity errors provided in each product. Where we have interpolated the velocity, we define the error as 10% of the estimated easting and northing velocity components in each timestamped pixel. We combine the easting and northing velocity errors normal to the flux gate through quadrature. Similarly, the thickness errors are defined as the sum through quadrature of the BedMachine bed elevation error where available and the error in the timestamped surface elevation estimate. We assume a 1 m error in the baseline REMA 200 m DEM, which is equivalent to the 90th percentile of the errors in the mosaic[97], and a 0.1 m yr$^{-1}$ error in the linear surface elevation change[66]. Following ref. [94], we define invalid thickness estimates as those that are <20 m where the ice is flowing faster than 100 m yr$^{-1}$, and we estimate the thickness in those pixels using a weak correlation ($R^2 = 0.55$) fit between the time-average velocity and the remaining valid baseline thickness pixels and we assume a 50% thickness error in these locations. Across the 'FG1' flux gate, this correction replaces 9.5% of all gate pixels and increases our average FG1 gate thickness by 26 m, resulting in an 1.2% increase in discharge.

In all cases throughout the manuscript, the presented discharge errors are defined as

$$\sigma D = \Sigma \left( (D_{max} - D) + (D - D_{min})/2 \right), \qquad (5)$$

where the pixel-based discharge errors are summed across all gate pixels falling within the defined basin. Therefore, we do not perform any mathematical reduction of the errors presented in the manuscript. Across the ASE as a whole, we obtain an average error of 14.9%. If instead we summed our pixel-based errors through quadrature, the average error would be 0.4%.

## Surface mass balance
We use SMB output from three regional climate models. We draw predominantly on the Regional Atmospheric Climate Model version 2.3p2 (RACMO2.3p2), which is forced at its lateral boundaries by ERA5 and run at $27 \times 27$ km resolution[54]. It has been especially adapted for use in the polar regions by adopting a multilayer snow model that includes routines for melt, percolation, refreezing, runoff, snow albedo, and drifting snow. RACMO2.3p2 has been extensively evaluated and compares favourably with observations of near-surface climate and SMB; in particular, RACMO2.3p2 SMB estimates are similar to and strongly correlated with radar-derived snow accumulation in the Getz and Thwaites Glacier basins[54].

These data are supplemented by SMB estimates from the Modèle Atmosphéric Régional (MAR)[56,57] and the Danish regional climate model HIRHAM5[55]. MAR is a hydrostatic model specifically designed for polar areas. It is forced at its boundaries by ERA5, runs at $35 \times 35$ km resolution, and includes routines for meltwater refreezing, snow metamorphism, snow surface albedo, and drifting snow[57]. MAR monthly-mean SMB output is available for 1980–2021. HIRHAM5 is a hydrostatic model forced at its boundaries by ERA-Interim. It is run at 0.11° resolution and includes a new albedo scheme[55], sublimation, surface melt, retention, refreezing, and runoff[103] but does not include wind redistribution of snow. HIRHAM5 daily-mean SMB output is available for 1980–2018.

Cumulative SMB anomalies are calculated as follows. In each spatial cell, we cumulatively sum timeseries of anomalies relative to the 1980–2008 climatological mean of that cell. We use 1980–2008 rather than the 1979–2008 mean so that we can use the same climatological period for all SMB datasets. These pixel-based cumulative SMB

anomalies are integrated across each glacier drainage basin to obtain regional cumulative SMB anomalies. SMB anomaly errors are taken as the root sum square of the annual errors[5,70], which we take as 14.8% of the annual SMB for all products[2].

## Mass balance

We linearly interpolate both the discharge and SMB timeseries to monthly intervals from January 1996 to December 2021. The mass balance is then the simple difference between these variables. The mass balance error is assumed to be the root sum square of the discharge and SMB errors. Cumulative mass change errors are defined as the root sum square of the annual mass balance errors, excluding errors in initial ice thickness.

## Impact of snowfall drought on historical mass balance

Annual grounding line discharge from the ASE is estimated by linearly interpolating the values presented in ref. [4] to annual intervals. Ref. [4] did not present grounding line discharge estimates for Getz. For the purpose of this analysis we estimate grounding line discharge from the Getz basin during 1974–1996 by scaling the annually interpolated values from ref. [4] according to the average relative contribution of the Getz basin to the total ASE discharge presented here since 1996. This approach assumes a synchronous response from all basins which is contrary to observations[9]. These discharge estimates are therefore used only to illustrate the potential impact of historical SMB changes on ASE-wide mass balance and are not a true estimate of historical ASE grounding line discharge.

## Atmospheric rivers

We use an existing ice-sheet wide, 3-hourly catalogue of ARs, derived using a Polar-specific AR detection algorithm[45], which is based on contiguous areas of very high (98th percentile) meridional component of vertically-integrated water vapour transport ($vIVT$). It is calculated using vertically-integrated specific humidity and meridional wind speeds from MERRA-2. From this catalogue, we include ARs only if they intersect the grounded ASE (basins G-H and F-G) and amalgamate AR detections if they are separated by less than 12 hours (accounting for any changes in AR footprint during that time). The precipitation provided by ARs is calculated by integrating daily RACMO2.3p2 precipitation within the landfalling-portion of the AR footprint from the start of the AR detection period and until 24 hours following the end of the event.

## Climate indices and historical snowfall drought

We use four climate indices to provide insight into the mechanistic drivers of potential historical periods of snowfall drought. The first of these indices is the Oceanic Niño Index (ONI) is a key indicator of the El Niño Southern Oscillation. It is a rolling 3-month average sea surface temperature anomaly (relative to the mean of the surrounding 30 years) in the east-central tropical Pacific between 120°–170°W (the Niño-3.4 region). An El Niño is considered to be present when the ONI is +0.5 or greater, whereas a La Niña is considered active when the ONI is −0.5 or lower. In Fig. 6, we shade sections of the time series only when the portion of the time-series between each zero crossing contains at least one value that exceeds these thresholds. The second climate index is the Tripole Index for the Interdecadal Pacific Oscillation[104] (IPOTPI), of which we use the HADISST1.1 filtered version. The IPOTPI is based on the difference between the sea surface temperature anomaly averaged over the central equatorial Pacific (10°S–10°N, 170°E–90°W) and the average of the sea surface temperature anomaly in the Northwest (25°N–45°N, 140°E–145°W) and Southwest Pacific (50°S–15°S, 150°E–160°W), with anomalies calculated relative to January 1971 to January 2000 period. We also use the Southern Annular Mode (SAM)[105], which is a station-based index defined as the zonal pressure difference between 40°S and 65°S.

Finally, we also use an ASL index, defined as the annual average of the 500-hPa geopotential height from 75°S to 60°S and 170°E to 70°W, normalised relative to the 1941–1990 period, derived from proxy-based climate reconstructions[80].

We define periods of snowfall drought (red and pink shading in Fig. 6) as periods of annual snow accumulation from ice cores or annual SMB from RACMO2.3p2 that are >1.5 median absolute deviations below the 1979–2009 median for at least two years.

## Ice-equivalent thickness change due to snowfall drought

The total mass change due to the 2009–2013 snowfall drought was -260 Gt. We estimate these-equivalent thickness change due to this mass change as follows. When distributed evenly over 1 m², 1 mm of water of density 1000 kg m⁻³ weights 1 kg. Therefore, the ice surface height change in water equivalent, $H_{w.e}$ due to the 260 Gt mass change, $M$, is

$$H_{w.e} = M/A, \qquad (6)$$

where $A$ is the total area over the grounded ASE basins (basins G-H and F-G), which we calculate as 550,680 km². The ice-equivalent height change is then simply given by $H_{w.e}.(\rho_w/\rho_i)$ where $\rho_w$ and $\rho_i$ are the densities of water (1000 kg m⁻³) and ice (917 kg m⁻³), respectively.

## Data availability

BedMachine version 2[95,96] is available from: https://nsidc.org/data/NSIDC-0756. The REMA DEM is available from: https://data.pgc.umn.edu/elev/dem/setsm/REMA/mosaic/v1.1/200m/. Rates of ice surface elevation change[66] are available from: https://digital.lib.washington.edu/researchworks/handle/1773/45388. The 'GL1' flux gate is available from: https://tc.copernicus.org/articles/12/521/2018/tc-12-521-2018-assets.html. MERRA-2 reanalysis data are available from: https://disc.gsfc.nasa.gov/. RACMO2.3p2 output is available from M. van den Broeke and J. M. van Wessem on request. Ice core data are available from: https://data.bas.ac.uk/full-record.php?id=GB/NERC/BAS/PDC/00940. The AR catalog is available from: https://www.earthsystemgrid.org/dataset/ucar.cgd.artmip.tier1.catalogues.wille_vIVT.html. The ONI data are available from (https://origin.cpc.ncep.noaa.gov/products/analysis_monitoring/ensostuff/ONI_v5.php), the IPOTPI data are available from https://psl.noaa.gov/data/timeseries/IPOTPI/ and the SAM data are available from https://climatedataguide.ucar.edu/climate-data/marshall-southern-annular-mode-sam-index-station-based. All data required to reproduce the results are available from: https://doi.org/10.5281/zenodo.7520043.

## Code availability

All code required to reproduce the results are available from: https://doi.org/10.5281/zenodo.7520043. Some of the analyses and figure preparation used functions in the Antarctic Mapping Tools for MATLAB[106].

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

## Acknowledgements

B.J.D. and A.E.H. are supported by the ESA Polar+ Ice Shelves project (ESA-IPL-POE-EF-cb-LE-2019-834), the ESA Polar+ SO-ICE project (ESA AO/1-10461/20/I-NB), which both are part of the ESA Polar Science Cluster, and the NERC DeCAdeS project (NE/T012757/1). Data processing was performed on ARC3 and ARC4, part of the High-Performance Computing facilities at the University of Leeds, UK. B.J.D. also thanks Nicolaj Hansen and Ruth Mottram for providing HIRHAM5 output, Christopher Kittel for providing MAR output, and Jonathan Wille for providing an updated atmospheric river dataset.

## Author contributions

B.J.D designed the study, performed the velocity measurements, conducted the analysis, and wrote the manuscript. A.E.H. supported the study design and velocity measurements. R.R. provided technical computing support. S.V. provided the firn air content estimate and J.M.v.W. provided the RACMO2.3p2 data. H.L.S. provided the model-optimized velocity estimate over Getz in 1996. A.E.H., S.V., J.M.v.W., M.R.v.d.B., P.R.H., H.L.S., and P.D. contributed to the manuscript preparation.

## Competing interests

The authors declare no competing interests
