## [Peer Review File · Nature Communications]

Sea level rise from West Antarctic mass loss significantly modified by large snowfall anomaliesREVIEWER COMMENTS

Reviewer #1 (Remarks to the Author):

Review of Davison et al.- Sea level rise from West Antarctic mass loss significantly modified by large snowfall anomalies

This paper examines a 25 year record of mass flux and mass balance data from the Amundsen Sea Embayment. The authors apportion the contributions of dynamic and surface mass balance (SMB) contributions to overall mass change, and identify anomalies in the SMB. They also investigate the causes of these SMB anomalies.

Their main conclusions are that while precipitation anomalies can have large short term influences on mass change in this region, they are small when compared to longer term changes.

Generally, the overall story could be clearer as the paper examines several difference aspects of mass change in this area, but the key finding presented in the title are only a small part of this analysis. I was left unsure if this is a paper describing mass changes in this area, or one specifically about anomalous precipitation events, so the focus does need to be made clearer throughout.

The analysis of the anomalous precipitation events are a new contribution to the field, and have the potential to be a useful and important result, but there are further steps needed in the analysis for this to be an impactful publication. The authors show that these precipitation anomalies can be important on short timescales. However, given that they argue these events have a very small contribution in the longer term, and spend some space arguing for the importance of oceanic influence in the future, it could be made clearer why the finding about precipitation is important. For example, the future frequency of these events, or the detailed atmospheric conditions mentioned that caused them, are not examined. If they are not important now, will they change in the future? Only atmospheric rivers are really examined, and determined not to be important in the future. While I appreciate the value of presenting scientific results that show things are not important instead of just focussing on the high-impact processes, this paper could do more to show why the reader should be interested in these events if they have no long term impact, and may not do so in the future.

Furthermore, result relies heavily on RACMO snowfall being correct. Although RACMO output may well be the closest we can get to the truth at this time, it is a big caveat of the results that the authors are assuming RACMO to be the truth so this should be made clearer.

Additionally, for the anomalous events to be a key result, the paper needs to explain in more detail the mechanism by which snowfall events relate to mass loss i.e. what proportion of snowfall is runoff if any? Could excess snowfall be melting and contributing to firn air depletion, having further impacts on mass loss in the future? Does the albedo of the ice shelf change with more or less fresh snow? The reader (I think) is expected to assume here that more snowfall = less mass loss and vice versa, but the story on the ice shelf surface is likely not that straightforward.

Comments on specific sections:

Line 128 (Mass balance)

The errors on mass balance are quite large- can these be compared with the changes caused by the precipitation anomalies (i.e. do these changes fall within the error bounds?). It's hard to see this from Fig 3b.

Line 148 (Anomalous precipitation)

How was the period of these events determined? i.e. looking at Fig 3b the SMB

contributions remains low beyond the 2013 cut-off, and the later high precipitation period is still a lower SMB than the early 2000s. Although you mention this was following work for Getz, can you be sure this low applies to the whole sector considered here? (While I understand your reasoning presented in the the paragraph beginning line 153, this is only really considering 5 year periods, why 5 years?).

Figure 3b

This figure contains the key story of the paper but is very difficult to interpret with the cut in the y axis. It may be preferable to present all data on the same axis, and have an additional 'zoomed in' plot to show the surface changes. The current plot makes comparing the surface with the dynamic extremely difficult, especially during the key period of the low snowfall anomaly discussed. It's actually doing a disservice to the results that you've found as it's not easy to pick out the change in SMB contribution during the anomalies examined.

Fig 5

This figure just has too much information. The number of very small plots makes it very hard to read, even when zoomed in e.g. the triangles in (j) are impossible to distinguish. Line 301

Is this assumption valid? Even if the low snowfall is uniform across the region (which seems unlikely) there are other processes going on to redistribute the snowfall that does occur (wind redistribution, surface melt etc).

Methods

The methods sections are generally clearly written and easy to follow. However, in terms of reproducibility more information needs to be given about the RACMO output provided and the model setup conditions used to provide the data for this study given it is such a key part of the findings. Information about this data is also not provided in the data availability section.

The choice of MERRA reanalysis here could also be better justified. Reanalysis products are notoriously unreliable in different ways over the polar regions, and more than one product could be compared to see if the same results are found if there is not a compelling reason for only using MERRA.

Reviewer #2 (Remarks to the Author):

General comments

This study takes a global look at the mass balance of the Amundsen Sea Embayment (ASE) and assesses how ice discharge along with annual precipitation variability including extreme precipitation events impacts the overall mass balance. The authors convincingly use a series of state-of-the-art datasets to clearly describe the grounding line ice drainage and velocity in the (ASE) and how that quantitatively affects sea-level rise. Then the authors effectively describe two different hydrological situations; a period of snow drought and a period of extreme precipitation via atmospheric rivers, and how they impact and mitigate the effects from the ice discharge thus creating a complete picture of mass balance in the ASE. The use of visually pleasing figures and clear presentation of results makes this a very robust manuscript. While describing the ice discharge and precipitation patterns on the ASE are useful, but already well-documented, the analysis of these two ideas combined makes this manuscript novel and a great addition to the established literature. The methodology is detailed and clearly described. I would be happy to see this manuscript published after one major comment in the Discussion section and a series of minor comments are addressed.

Major comments

Lines 315-334: I do appreciate that the authors placed caveats about the interpretation of future AR effects, but I have quite a few issues with these back-of-the-envelope calculations.

1. ARs are sensitive to the method of detection. For instance, the AR detection algorithm in Wille et al., 2021 only detects ARs if the vIVT exceeds the 98th percentile which is a relatively high threshold compared to other AR detection methods. The authors should mention that the method of AR detection introduces uncertainty into the calculation.
2. ARs have a very high degree of inter-annual variability so the trend calculation is sensitive to the chosen set of years along with the season of the trend. Plus the trends are sensitive to the reanalysis dataset chosen for the AR detection. For example, Wille et al., 2021 showed that the AR detection algorithm had noticeable difference in trends between MERRA-2 and ERA-5 and that the trends changed a lot between different seasons which has implications for future snowfall attribution. Also there is no reason to believe that the observed trends from 1980-2018 will continue in a linear fashion until the year 2100. Therefore, I don't believe it is accurate to say ARs will occur on average 14-18 days per year by 2100. I suggest removing this result as we really would need a future climate projection focused on AR detections to provide good insight.
3. The future precipitation estimations are also based on unreliable assumptions. While it is reasonable to expect future AR intensity to increase given the greater atmospheric water-vapor content, the actual amount of extra precipitation is more complicated. Firstly, AR precipitation does not linearly correlate to AR intensity. In fact, Wille et al., 2021 and Maclennan et al., 2022 show that precipitation responds in an almost exponential way to increasing IVT (This detail also should be mentioned in the Results section). However, AR related precipitation does appear to respond linearly to increased AR frequency. Anyway, I suggest removing this calculation as well.
4. I do generally agree that ARs in the ASE region will not do much to mitigate sea level rise and don't have an issue with this claim even if the calculations going into this claim are not reliable. But the more important question is how future changes in AR activity and intensity over the East Antarctic will mitigate the sea level rise from ice discharge in the ASE. I think it is reasonable to say that the ice discharge in ASE is too great to be mitigated by future AR changes even if they produce much more snowfall, but what happens in the East Antarctic is more important.
5. Also the authors could benefit from comparing the future ASE to present day Greenland where ARs do cause substantial snowfall, but even more surface melting. See Mattingly et al., 2018.

Minor comments

Line 13: Draining "into" the...

Line 14: on "the" ASE

Line 15: Mention that this is a satellite record.

Line 22: The double negative is a bit confusing.

Figure 1: Nice figure design, but mention that the color bar is scaled logarithmically in the figure caption.

Line 42: The word "filling" is a bit unclear here. Please briefly mention how the pressure and position of the Amundsen Sea Low changes in relation to ENSO.

Line 44: Overall, a good description on atmospheric influences on basal melting. The authors should also discuss how the Southern Annular Mode impacts basal melting in the region. See Verfaillie et al., 2022

Results: The results here are very substantial and methodically explained. The only thing potentially worth adding is a discussion on the seasonality of the SMB on the ASE. Please discuss which season has the greatest contribution to snowfall and which seasons during the study period had the greatest anomalies of SMB in terms of magnitude. This would benefit the analysis of Figure 4 which only shows relative

seasonal precipitation anomalies. Also surface melt should be briefly discussed in reference to present SMB even if it is a minor contribution. See Adusumilli et al., 2021.

Figure 2 and Figure 3, Supplementary Figure 1, Supplementary Figure 10a: These figures like most figures in this manuscript are very well designed but consider plotting every other year on the x-axis. The years look a bit crowded.

Figure 3a: Please describe what the gray floating bar refers to.

Figure 4d, f, h, j: It's a bit difficult to discern the surface air pressure anomalies in this panels. Perhaps consider reducing the contour and wind barb frequency to make a cleaner figure.

Line 157: It might be worth mentioning that atmospheric river activity in the WAIS region was relatively low during the 2009-2013 period and helps explain why those years had anomalously low SMB. See Figure 1 in Wille et al., 2019.

Supplementary Figure 3c: Please explain in the figure caption how the cumulative anomaly is calculated.

Line 194: Very impressive to see extreme precipitation events explains over half the winter precipitation in 2020. Just saying this is a cool result, no revision needed.

Supplementary Figure 6: I don't quite understand the red line in this figure. How is the mass anomaly related to the daily snowfall on the x-axis and how does this relate to the faint gray lines? Please change the figure or clarify the figure caption.

Line 206: Please cite Wille et al., 2021 as it discusses the atmospheric pressure dipole as a key component of landfalling atmospheric rivers across Antarctica.

Line 247: Perhaps clarify that these numbers before the comma are related to mass loss.

Line 264: Include abbreviation of "Southern Annular Mode" after its mention.

Line 266: The authors should cite literature that supports the idea of a positive SAM deepening the ASL.

Line 266-267: It wasn't clear that the authors were referring to the hypothetical situation of the ASL here. Like change the sentence to, "which theoretically would have acted to deepen the ASL. However, we observed the opposite ..."

Figure 6b: If I'm reading this panel correcting, I don't understand how the cumulative SMB anomaly is always positive.

Figure 6: Consider changing the red and pink bars to colors that are more colorblind friendly. Even I'm finding the colors a little confusing.

Line 271: This is repeating the same result on line 248. In fact, on line 248, the authors mention the total 2009-2013 loss was 202 Gt per year but on line 272, the authors mention the total mass change was 260 Gt per year (which is also unclear whether it is positive or negative). Which total mass loss number is correct here?

Line 318: We don't have great resolution on future AR projections in the Amundsen and Bellingshausen yet. Perhaps mention that the increasing trend is expected to be broadly observed across the Southern Ocean

Line 349: It could be interesting to conclude here that periods of increased snowfall and SMB are not enough to meaningfully mitigate the sea-level rise contribution from the relatively recent increases in ice discharge.

Methods: Can the authors add a few sentences in the Methods section detailing the RACMO model and discussing any potential biases?

Line 467-469: The version of the AR detection algorithm used in this study uses the v-component of integrated vapor transport (vIVT). So the authors can just say "... it is based on contiguous areas of very high (98th percentile) v-component of vertically integrated water vapour transport (vIVT) from MERRA-2."

Line 488: It's Southern Annular Mode not Southern Annual Mode.

References:

Verfaillie, D. et al. The circum-Antarctic ice-shelves respond to a more positive Southern Annular Mode with regionally varied melting. *Communications Earth & Environment* 3, 139 (2022).

Adusumilli, S., A. Fish, M., Fricker, H. A. & Medley, B. Atmospheric River Precipitation Contributed to Rapid Increases in Surface Height of the West Antarctic Ice Sheet in 2019. *Geophysical Research Letters* 48, e2020GL091076 (2021).

Wille, J. D. et al. West Antarctic surface melt triggered by atmospheric rivers. *Nat. Geosci.* 12, 911–916 (2019).

Wille, J. D. et al. Antarctic Atmospheric River Climatology and Precipitation Impacts. *Journal of Geophysical Research: Atmospheres* 126, e2020JD033788 (2021).

Mattingly, K. S., Mote, T. L. & Fettweis, X. Atmospheric River Impacts on Greenland Ice Sheet Surface Mass Balance. *Journal of Geophysical Research: Atmospheres* 123, 8538–8560 (2018).

Response to reviews of: “Sea level rise from West Antarctic mass loss significantly modified by large snowfall anomalies”

Note that all line numbers refer to the tracked changes version of the revised manuscript.

Reviewer #1

ID	Reviewer’s comment	Response
1	This paper examines a 25 year record of mass flux and mass balance data from the Amundsen Sea Embayment. The authors apportion the contributions of dynamic and surface mass balance (SMB) contributions to overall mass change, and identify anomalies in the SMB. They also investigate the causes of these SMB anomalies. Their main conclusions are that while precipitation anomalies can have large short term influences on mass change in this region, they are small when compared to longer term changes.	We thank the reviewer for investing their time in providing a thorough and detailed review of our manuscript. We have revised the manuscript according to each of the reviewer’s comments, which we think has greatly improved the manuscript. We provide our responses to the reviewer’s overarching comments here, and to each of the minor comments in turn below.
2	Generally, the overall story could be clearer as the paper examines several difference aspects of mass change in this area, but the key finding presented in the title are only a small part of this analysis. I was left unsure if this is a paper describing mass changes in this area, or one specifically about anomalous precipitation events, so the focus does need to be made clearer throughout.	Done. The paper is focused primarily on the anomalous precipitation events – their characteristics, impacts on mass balance during the study period and associated atmospheric conditions. We have made several modifications to the manuscript to clarify this. However, this focus requires that we also calculate and present a full time-series of discharge and mass balance, in order to put the precipitation events in context. By comparing these time-series of SMB, discharge and mass balance, we show that (1) the vast majority of the SMB contribution to SLR is due to the two extreme events that we characterise (Lines 162-163), and (2) these events significantly modify the mass balance of the ASE over 2-5 year time-periods, such that annual mass balance has not scaled with grounding line discharge as many researchers might expect (Lines 393-396). These are important findings that were not previously known or appreciated in the literature, and we have clarified this point throughout the revised manuscript. For example, on lines 136-139, we clarify that changes in discharge are not mirrored by changes in mass balance because of large variations in SMB over a range of time-scales. In addition, on lines 396-400, we emphasise that whilst discharge has increased progressively, mass balance has been comparatively steady since 2010 because of these anomalous precipitation events, so they do have an important impact on the sea level contribution from the region. Also see lines 147-148, 266-267, 272-274, 298-307 and 323-325. However, given the large increases in grounding line discharge in the region, which is the primary driver of total mass loss, we feel it would be remiss to not emphasise the importance of discharge and encourage further research into the processes that lead to changes in discharge. Finally, given that the paper is targeted at a ‘letters’ journal, we intend the title to highlight the main conclusion rather than describe all of the analysis or results.

3	The analysis of the anomalous precipitation events are a new contribution to the field, and have the potential to be a useful and important result, but there are further steps needed in the analysis for this to be an impactful publication. The authors show that these precipitation anomalies can be important on short timescales. However, given that they argue these events have a very small contribution in the longer term, and spend some space arguing for the importance of oceanic influence in the future, it could be made clearer why the finding about precipitation is important. For example, the future frequency of these events, or the detailed atmospheric conditions mentioned that caused them, are not examined. If they are not important now, will they change in the future? Only atmospheric rivers are really examined, and determined not to be important in the future. While I appreciate the value of presenting scientific results that show things are not important instead of just focussing on the high-impact processes, this paper could do more to show why the reader should be interested in these events if they have no long term impact, and may not do so in the future.	Done. Based on the major comments from reviewer #2, we have significantly revised our discussion of the potential future importance of ARs. We have also placed that discussion in the context of future SMB simulations, which show large increases in future SMB that in some simulations do lead to overall mass gain at the ice sheet scale. We choose to focus on historical periods of snowfall drought rather than future periods, because their potential to impact ice sheet dynamics is theoretically greater during times of quasi-mass balance and because SMB is projected to increase in future, so droughts may become less frequent or extreme. We argue that this does not diminish their importance: based on our analysis presented in Figure 6, it is plausible that snowfall droughts have likely occurred during the 20th century and we suggest that these could have contributed to the initial mass imbalance during the 20th century from these glaciers that have, to the best of our knowledge, precipitated the ongoing mass loss that we observe today. (Lines 323-325). This is an important result because it adds nuance and depth to the existing literature that generally focuses on the impact of historical and ongoing ocean forcing as the sole driver of ice mass loss from this region - this paper will, therefore, require many researchers to revise their perspective regarding the role of SMB variations in driving ASE mass change, and we suspect it will stimulate further research into atmospheric rivers and droughts in this region as well as across the broader ice sheet. Finally, our analysis of the impact of anomalous precipitation on mass balance variations highlights the different time-scales of variability of discharge and SMB, which means that rates of mass loss are highly variable in time and do not necessarily scale with grounding line discharge. We note therefore (lines 399-400) that this means that it is not possible to characterise mass loss from short-term observations, so sustained monitoring of ice sheet mass balance is necessary.
4	Furthermore, result relies heavily on RACMO snowfall being correct. Although RACMO output may well be the closest we can get to the truth at this time, it is a big caveat of the results that the authors are assuming RACMO to be the truth so this should be made clearer.	Done. Thirdly, regarding the reliance on RACMO. In the revised manuscript, we have incorporated output of two additional regional climate models (HIRHAM and MAR) to provide further clarity on the SMB and precipitation estimates. There are some differences between these models. Notably, HIRHAM shows greater SMB anomalies after 2010 relative to MAR and RACMO (see Sup. Fig. 3). Overall, however, the models agree in terms of the timing, magnitude and spatial distribution of SMB anomalies (see Sup. Fig. 5 and 7). In addition, the differences between mass balance estimates utilising each of the SMB models are small – much smaller than the errors in the mass balance – and do not alter our conclusions (see Sup. Fig. 2). In the same vein, and in response to one of the reviewer’s comments below (comment #11), we have also incorporated a second atmospheric reanalysis dataset (ERA5), which further supports our findings.
5	Additionally, for the anomalous events to be a key result, the paper needs to explain in more detail the mechanism by which snowfall events relate to mass loss i.e. what proportion of snowfall is runoff if any? Could excess snowfall be	Done. Mass changes at the surface are driven by net accumulation (SMB), so in this study we use SMB rather than snowfall. Given that the regional climate models account for runoff and refreezing in their SMB calculation, it is not necessary to further account for them in our mass balance analysis. However, we note that SMB of the AIS is

	melting and contributing to firm air depletion, having further impacts on mass loss in the future? Does the albedo of the ice shelf change with more or less fresh snow? The reader (I think) is expected to assume here that more snowfall = less mass loss and vice versa, but the story on the ice shelf surface is likely not that straightforward.	in first order defined by snowfall, with sublimation being an order of magnitude smaller and runoff essentially zero in the contemporary AIS climate (Van Wessem and others, 2018). We have clarified this point in the manuscript (Line 162), where we show that precipitation is the dominant contributor to SMB in our study region. This paper is focused on the grounded ice sheet in the Amundsen Sea Embayment of West Antarctica, therefore, change in the ice shelf surface properties through albedo or other mechanisms is out of scope of the study. On the ice sheet in the ASE, the reader would be correct to infer from our results that more snowfall will reduce ice mass loss from the region – recourse to further consequences, such as changes in firm air content following subsequent snowpack melting, are not necessary to understand the observed mass changes (Lines 163-166). As stated above, melt volumes are small and the vast majority of meltwater is thought to refreeze in the firm, so that mass loss through runoff is essentially zero. In any case, these will be captured by the subsequent time-steps in the regional climate models and therefore feed into our mass balance estimates. We were surprised and interested to see from our results that the cumulative mass input from several large precipitation events was large enough to notably change the mass balance of this region, and we think the scientific community will find that result equally interesting.
6	Line 128 (Mass balance) The errors on mass balance are quite large- can these be compared with the changes caused by the precipitation anomalies (i.e. do these changes fall within the error bounds?). It's hard to see this from Fig 3b.	Done. We have addressed this comment by including error estimates for the mass change during each of the extreme precipitation events. For the 2009-2013 period of snowfall drought, the mass change due to anomalously low SMB is more than three times the error in the overall mass change during the same period. For the 2019-2020 period of high snowfall, the mass change due to anomalously high SMB is similar to the overall mass change error during the same period, but we note that the errors in the SMB component during each time period are much smaller.
7	Line 148 (Anomalous precipitation) How was the period of these events determined? i.e. looking at Fig 3b the SMB contributions remains low beyond the 2013 cut-off, and the later high precipitation period is still a lower SMB than the early 2000s. Although you mention this was following work for Getz, can you be sure this low applies to the whole sector considered here? (While I understand your reasoning presented in the paragraph beginning line 153, this is only really considering 5 year periods, why 5 years?).	Done. The events were determined based on the gradient in the cumulative SMB anomaly, which indicates whether SMB is above or below the climatological mean. Whereas the absolute cumulative anomaly depends on the preceding anomalies. For example, after 2013, the reviewer is correct that the cumulative SMB anomaly remains low, but it is not changing much on annual timescales because SMB is close to the climatological mean. We have clarified this point in the manuscript (Lines 159-166).
8	Figure 3b This figure contains the key story of the paper but is very difficult to interpret with the cut in the y axis. It may be preferable to present all data on the same axis, and have an additional 'zoomed in' plot to show the surface changes. The current plot makes comparing the	Comment. We experimented with this figure as the reviewer suggested. However, plotted without the break in the y-axis, it is very hard to discern the changes in SMB on which the paper focuses. We also feel that removing the axis cut does not improve the comparisons between dynamic, surface and total mass change, therefore, we have chosen to retain the original design of this figure.

	surface with the dynamic extremely difficult, especially during the key period of the low snowfall anomaly discussed. It's actually doing a disservice to the results that you've found as it's not easy to pick out the change in SMB contribution during the anomalies examined.	
9	Fig 5 This figure just has too much information. The number of very small plots makes it very hard to read, even when zoomed in e.g. the triangles in (j) are impossible to distinguish.	Done. The reviewer is correct that Figure 5 contains a lot of detail, however it does allow the reader to assess all the information relating to the 2019-20 precipitation anomalies in one location. In response to this comment, we have made some modifications to the figure to improve readability. Firstly, we have adjusted the size and position of the triangles in (d) so that they are easier to discern. Secondly, we have adjusted the contours and wind barbs in panels (f) and (g), so those figures are much clearer. We have also grouped the panels according to the theme of information they present (precipitation, winds and atmospheric rivers). We think that these changes have greatly improved the readability of the figure and that all panels are necessary to characterise the extreme precipitation events we discuss in the paper.
10	Line 301 Is this assumption valid? Even if the low snowfall is uniform across the region (which seems unlikely) there are other processes going on to redistribute the snowfall that does occur (wind redistribution, surface melt etc).	Done. The reviewer is correct that the real change in height due to the snowfall anomaly will be different and much more spatially variable than this. In practice, the relative snowfall anomaly is more negative near the coast and less negative further up-glacier. To quantify this, we integrated the SMB anomaly on a grid-cell basis to visualise the changes in height due to the drought (since we use SMB, wind redistribution and surface melt are accounted for) – see new Supplementary Fig 11. Even with this more sophisticated approach, the relative thinning is generally less than 1 m close to the coast, reaches over 1.5 m in parts of the Getz basin and decreases quickly to less than 0.5 m up-glacier. We have modified the text to reflect this new analysis (Lines 331-343).
11	Methods The methods sections are generally clearly written and easy to follow. However, in terms of reproducibility more information needs to be given about the RACMO output provided and the model setup conditions used to provide the data for this study given it is such a key part of the findings. Information about this data is also not provided in the data availability section. The choice of MERRA reanalysis here could also be better justified. Reanalysis products are notoriously unreliable in different ways over the polar regions, and more than one product could be compared to see if the same results are found if there is not a compelling reason for only using MERRA.	Done. We now include a description of the key components of RACMO used and have updated the data availability statement to reflect this. The relevant text in the methods is: “We use SMB output from three regional climate models. We draw predominantly on the Regional Atmospheric Climate Model version 2.3p2 (RACMO2.3p2), which is forced at its lateral boundaries by ERA5 and run at 27x27 km resolution⁵⁵. It has been especially adapted for use in the polar regions by adopting a multilayer snow model that includes routines for melt, percolation, refreezing, runoff, snow albedo and drifting snow. RACMO2.3p2 has been extensively evaluated and compares favourably with observations of near-surface climate and SMB; in particular, RACMO2.3p2 SMB estimates are similar to and strongly correlated with radar-derived snow accumulation in the Getz and Thwaites Glacier basins⁵⁵.” Note that we now also draw on MAR and HIRHAM to corroborate our findings using RACMO output.

		In response to the second part of the reviewer's comment, we have incorporated ERA5 into our analysis and found that it agrees well with MERRA-2 during the extreme events. This provides evidence that the conclusions from our paper are robust to our choice of global reanalysis dataset.
--	--	--

Reviewer #2

General comments

This study takes a global look at the mass balance of the Amundsen Sea Embayment (ASE) and assesses how ice discharge along with annual precipitation variability including extreme precipitation events impacts the overall mass balance. The authors convincingly use a series of state-of-the-art datasets to clearly describe the grounding line ice drainage and velocity in the (ASE) and how that quantitatively affects sea-level rise. Then the authors effectively describe two different hydrological situations; a period of snow drought and a period of extreme precipitation via atmospheric rivers, and how they impact and mitigate the effects from the ice discharge thus creating a complete picture of mass balance in the ASE. The use of visually pleasing figures and clear presentation of results makes this a very robust manuscript. While describing the ice discharge and precipitation patterns on the ASE are useful, but already well-documented, the analysis of these two ideas combined makes this manuscript novel and a great addition to the established literature. The methodology is detailed and clearly described. I would be happy to see this manuscript published after one major comment in the Discussion section and a series of minor comments are addressed.

We thank the reviewer for investing their time in improving our manuscript by providing a comprehensive and considered review. We have been through each of the reviewer's comments in turn and modified the manuscript and figures accordingly, which we feel has greatly improved the manuscript. Our response to each comment is provided below.

Major comments

Lines 315-334: I do appreciate that the authors placed caveats about the interpretation of future AR effects, but I have quite a few issues with these back-of-the-envelope calculations.

1. ARs are sensitive to the method of detection. For instance, the AR detection algorithm in Wille et al., 2021 only detects ARs if the vIVT exceeds the 98th percentile which is a relatively high threshold compared to other AR detection methods. The authors should mention that the method of AR detection introduces uncertainty into the calculation.
2. ARs have a very high degree of inter-annual variability so the trend calculation is sensitive to the chosen set of years along with the season of the trend. Plus the trends are sensitive to the reanalysis dataset chosen for the AR detection. For example, Wille et al., 2021 showed that the AR detection algorithm had noticeable difference in trends between MERRA-2 and ERA-5 and that the trends changed a lot between different seasons which has implications for future snowfall attribution. Also there is no reason to believe that the observed trends from 1980-2018 will continue in a linear fashion until the year 2100. Therefore, I don't believe it is accurate to say ARs will occur on average 14-18 days per year by 2100. I suggest removing this result as we really would need a future climate projection focused on AR detections to provide good insight.
3. The future precipitation estimations are also based on unreliable assumptions. While it is reasonable to expect future AR intensity to increase given the greater atmospheric water-vapor content, the actual amount of extra precipitation is more complicated. Firstly, AR precipitation does not linearly correlate to AR intensity. In fact, Wille et al., 2021 and Maclennan et al., 2022 show that precipitation responds in an almost exponential way to increasing IVT (This detail also should be mentioned in the Results section). However, AR related precipitation does appear to respond linearly to increased AR frequency. Anyway, I suggest removing this calculation as well.
4. I do generally agree that ARs in the ASE region will not do much to mitigate sea level rise and don't have an issue with this claim even if the calculations going into this claim are not reliable. But the more important question is how future changes in AR activity and intensity over the East Antarctic will mitigate the sea level rise from ice discharge in the ASE. I think it is reasonable to say that the ice discharge in ASE is too great to be mitigated by future AR changes even if they produce much more snowfall, but what happens in the East Antarctic is more important.
5. Also the authors could benefit from comparing the future ASE to present day Greenland where ARs do cause substantial snowfall, but even more surface melting. See Mattingly et al., 2018.

We thank the reviewer for the thorough and considered comment on our discussion of future AR snowfall provision to the ASE. In response, we have completely rewritten this paragraph, opting not to present those back-of-the-envelope calculations (because of the large uncertainties the reviewer mentions) and instead focus

on providing a summary of potential future impacts on SMB from ARs and AIS-wide SMB projections, including a comparison with present-day Greenland, all of which we think is a more useful discussion for the manuscript than the previous version. (see lines 353-373).

ID	Reviewer's comment	Response
1	Line 13: Draining "into" the...	Done
2	Line 14: on "the" ASE	Done
3	Line 15: Mention that this is a satellite record.	Done. We now state that it is a "25-year (1996-2021) record of input-output mass balance...", which accounts for the fact that we use satellites for the grounding line discharge and models for the SMB.
4	Line 22: The double negative is a bit confusing.	Comment. We appreciate that it is confusing, however we couldn't find a better way to word this. For example "partially offsetting the total loss" is a bit misleading because the SMB is already included in the mass change figure and otherwise just repeats information earlier in the sentence.
5	Figure 1: Nice figure design, but mention that the color bar is scaled logarithmically in the figure caption.	Done
6	Line 44: Overall, a good description on atmospheric influences on basal melting. The authors should also discuss how the Southern Annular Mode impacts basal melting in the region. See Verfaillie et al., 2022	Done. We thank the reviewer for the feedback and suggestion. We have added a description of the impact of the Southern Annular Mode on basal melting in the region. "While influential on the wider Antarctic climate, the Southern Annular Mode (SAM) does not significantly influence winds directly over the Amundsen Sea, or rates of ocean melting at the grounding lines of the ASE ice streams that are rapidly thinning ^{18,33} " (lines 46-48)
7	Results: The results here are very substantial and methodically explained. The only thing potentially worth adding is a discussion on the seasonality of the SMB on the ASE. Please discuss which season has the greatest contribution to snowfall and which seasons during the study period had the greatest anomalies of SMB in terms of magnitude. This would benefit the analysis of Figure 4 which only shows relative seasonal precipitation anomalies. Also surface melt should be briefly discussed in reference to present SMB even if it is a minor contribution. See Adusumilli et al., 2021.	Done. Regarding the seasonality in SMB and snowfall: we note that Fig. 4a does show a time-series of absolute snowfall during each month, which shows that snowfall is lowest during the summer months. Given that the greatest relative anomalies in snowfall also occurred during seasons of greatest absolute snowfall, then it follows that the greatest absolute snowfall anomalies would also be in those seasons. We now include this detail on lines 189-191. Regarding the contribution of surface melt to SMB; here we adopt the SMB definition that includes runoff from the firn layer. Runoff is negligible, so it is safe to state that the contribution of surface melt (and sublimation) to SMB are very small in the ASE. So we do not think it needs to be discussed in the manuscript. Instead, we have justified our focus on precipitation by including the following text: "In the ASE, precipitation is the dominant contributor to SMB (it explains over 98% over the variance in SMB), with the remainder caused by sublimation. Runoff from the grounded ice sheet is essentially zero, as meltwater refreezes in the cold firn. Consequently, we focus in this study on investigating precipitation variations." (Lines 163-166)
8	Figure 2 and Figure 3, Supplementary Figure 1, Supplementary Figure 10a:	Done

	These figures like most figures in this manuscript are very well designed but consider plotting every other year on the x-axis. The years look a bit crowded.	
9	Figure 3a: Please describe what the gray floating bar refers to.	Done. (the ASE total mass change error)
10	Figure 4d, f, h, j: It's a bit difficult to discern the surface air pressure anomalies in this panels. Perhaps consider reducing the contour and wind barb frequency to make a cleaner figure.	Done. We have modified the figure as the reviewer suggested and were very pleasantly surprised by how much clearer the figure was as a result. Thanks very much for the suggestion.
11	Line 157: It might be worth mentioning that atmospheric river activity in the WAIS region was relatively low during the 2009-2013 period and helps explain why those years had anomalously low SMB. See Figure 1 in Wille et al., 2019.	Done. We thank the reviewer for pointing this out. We have added, "the precipitation deficit was caused by a reduced number of large (~1.5-2.5 Gt) precipitation events during 2009-2013, compared to all other 5-year periods during 1979-2021 (Fig. 4b; Sup. Fig. 4), consistent with fewer inferred AR events during this time period ⁴⁵ ..." (Lines 173-174)
12	Supplementary Figure 3c: Please explain in the figure caption how the cumulative anomaly is calculated.	Done. We now describe this more clearly in the methods. (Lines 520-525)
13	Line 194: Very impressive to see extreme precipitation events explains over half the winter precipitation in 2020. Just saying this is a cool result, no revision needed.	Comment. Thanks! We agree this is a neat result.
14	Supplementary Figure 6: I don't quite understand the red line in this figure. How is the mass anomaly related to the daily snowfall on the x-axis and how does this relate to the faint gray lines? Please change the figure or clarify the figure caption.	Done. We have simplified this figure (and the equivalent in the main text). We now only plot one histogram, which shows the change in number of daily event sizes during the period of interest compared to the rest of the time-series, and the red line of mass change. The mass change on the second y-axis is essentially the magnitude of the precipitation event size bin multiplied by the change in number of events. We think it is a useful addition because it emphasises that a small change in the number of large events can have a disproportionate impact on the mass of precipitation provided.
15	Line 206: Please cite Wille et al., 2021 as it discusses the atmospheric pressure dipole as a key component of landfalling atmospheric rivers across Antarctica.	Done. We already cite this paper at the appropriate point on this line.
16	Line 247: Perhaps clarify that these numbers before the comma are related to mass loss.	Done. This is now written as "The cumulative mass loss due to this snowfall drought was ~260 Gt or 51.4 Gt yr ⁻¹ (Fig. 3b), contributing positively to the total mass loss of 195 Gt yr ⁻¹ ."
17	Line 264: Include abbreviation of "Southern Annular Mode" after its mention.	Done

18	Line 266: The authors should cite literature that supports the idea of a positive SAM deepening the ASL.	Done. We now cite Turner et al. (2013) and Raphael et al (2016).
19	Line 266-267: It wasn't clear that the authors were referring to the hypothetical situation of the ASL here. Like change the sentence to, "which theoretically would have acted to deepen the ASL. However, we observed the opposite ..."	Done
20	Figure 6b: If I'm reading this panel correcting, I don't understand how the cumulative SMB anomaly is always positive.	Done. The reviewer is correct that the cumulative SMB anomaly cannot by definition always be positive. However, in this panel, we have normalised each timeseries between 0 and 1 to show relative changes in cumulative SMB anomaly in order to facilitate comparison between different basins. We have clarified this in the figure caption.
21	Figure 6: Consider changing the red and pink bars to colors that are more colorblind friendly. Even I'm finding the colors a little confusing.	Done
22	Line 271: This is repeating the same result on line 248. In fact, on line 248, the authors mention the total 2009-2013 loss was 202 Gt per year but on line 272, the authors mention the total mass change was 260 Gt per year (which is also unclear whether it is positive or negative). Which total mass loss number is correct here?	Done. Our wording here was not very precise and made it easy to confuse total precipitation anomalies and total mass change of the grounded ice. This is now written as: "The snowfall drought during 2009-2013 caused 51.4 ± 4 Gt yr ⁻¹ relative mass loss in those years, accumulating to a total SMB anomaly of -252.6 ± 81 Gt over the 5-year period (Fig. 3b)", (Lines 298-299) so it can be more easily distinguished from the total mass change over grounded ice of 195 Gt yr ⁻¹ mentioned earlier in the manuscript. Note that we have slightly revised our mass change estimates after incorporating multiple SMB estimates.
23	Line 318: We don't have great resolution on future AR projections in the Amundsen and Bellingshausen yet. Perhaps mention that the increasing trend is expected to be broadly observed across the Southern Ocean	Done. This paragraph has been rewritten following the major comment provided by the reviewer.
24	Line 349: It could be interesting to conclude here that periods of increased snowfall and SMB are not enough to meaningfully mitigate the sea-level rise contribution from the relatively recent increases in ice discharge.	Done. We thank the reviewer for the suggestion. We have reworded this section to: " Therefore, whilst discharge has progressively increased during the study period, the mass balance of the ASE has been comparatively steady since 2010. Thus, periods of anomalous precipitation do have a non-negligible impact on the sea level contribution from the ASE and estimates of mass balance over short time periods (5 years or less) may not be representative of longer-term mass changes." (Lines 396-400)
25	Methods: Can the authors add a few sentences in the Methods section detailing the RACMO model and discussing any potential biases?	Done. We now use SMB and precipitation output from three regional climate models and detail each in the methods, with specific focus on the performance of RACMO with respect to observations (see lines 503-525).
26	Line 467-469: The version of the AR detection algorithm used in this study	Done

	uses the v-component of integrated vapor transport (vIVT). So the authors can just say "... it is based on contiguous areas of very high (98th percentile) v-component of vertically integrated water vapour transport (vIVT) from MERRA-2."	
27	Line 488: It's Southern Annular Mode not Southern Annual Mode.	Corrected

REVIEWERS' COMMENTS

Reviewer #1 (Remarks to the Author):

In response to my previous review I'd like to thank the authors for their thorough response to my comments. The new version of the manuscript reads more clearly and on the basis of these changes I'd be happy to recommend it for publication.

Reviewer #2 (Remarks to the Author):

The authors have addressed my comments in a satisfying manner. I recommend this manuscript for publication. Congrats on the good work and I look forward to seeing this published.

I found a minor mistake when looking over the manuscript
Line 474: Change to "vIVT" method
I recommend a proofread for any other small mistakes

REVIEWERS' COMMENTS

Reviewer #1 (Remarks to the Author):

In response to my previous review I'd like to thank the authors for their thorough response to my comments. The new version of the manuscript reads more clearly and on the basis of these changes I'd be happy to recommend it for publication.

We are very happy that the reviewer is happy with our response to their previous review. We would also like to thank the reviewer again for taking the time to provide a thorough and constructive review of the manuscript, which greatly improved the manuscript.

Reviewer #2 (Remarks to the Author):

The authors have addressed my comments in a satisfying manner. I recommend this manuscript for publication. Congrats on the good work and I look forward to seeing this published.

We are very happy that the reviewer is satisfied with our response to their previous review and are grateful for their kind feedback in this review. We would like to thank the reviewer again for taking the time to provide a thorough and constructive review of the manuscript, which greatly improved the manuscript.

I found a minor mistake when looking over the manuscript

Line 474: Change to "vIVT" method – Done.

I recommend a proofread for any other small mistakes

We have proofread the manuscript and found the following small mistakes, which have been corrected:

- Line 30: "the ASE glaciers have..."
- Line 92: removed "rate"
- Line 107: replaced "data" with "observations"
- Line 133: deleted superfluous "%"
- Line 137: removed unnecessary comma
- Line 143: changed "from" to "of" and unnecessary commas
- Line 183: added space after "spring"
- Line 202: added "therefore"
- Line 224: removed unnecessary comma
- Line 247: removed "observed" and added comma after "ASE"
- Line 301: changed "timeseries" to "time-series"
- Figure 6 caption: RACMO2.3pt is now RACMO2.3p2
- Removed uppercase letters from subheadings